# SCoT: Unifying Consistency Models and Rectified Flows via Straight-Consistent Trajectories

**Zhangkai Wu**
School of Computing
Macquarie University
`zhangkai.wu@mq.edu.au`

**Xuhui Fan**[*]
School of Computing
Macquarie University
`xuhui.fan@mq.edu.au`

**Hongyu Wu**
School of Computing
Macquarie University
`hongyu.wu@students.mq.edu.au`

**Longbing Cao**
School of Computing
Macquarie University
`longbing.cao@mq.edu.au`

## Abstract

Pre-trained diffusion models are commonly used to generate clean data (e.g., images) from random noises, effectively forming pairs of noises and corresponding clean images. Distillation on these pre-trained models can be viewed as the process of constructing advanced trajectories within the pair to accelerate sampling. For instance, consistency model distillation develops consistent projection functions to regulate trajectories, although sampling efficiency remains a concern. Rectified flow method enforces straight trajectories to enable faster sampling, yet relies on numerical ODE solvers, which may introduce approximation errors. In this work, we bridge the gap between the consistency model and the rectified flow method by proposing a Straight-Consistent Trajectories (SCoT) model. SCoT enjoys the benefits of both approaches for fast sampling, producing trajectories with consistent and straight properties simultaneously. These dual properties are strategically balanced by targeting two critical objectives: (1) regulating the gradient of SCoT's mapping function to a constant and (2) ensuring trajectory consistency. Extensive experimental results demonstrate the effectiveness and efficiency of SCoT.

## 1 Introduction

Pre-trained diffusion models Ho et al. (2020); Song et al. (2021b); Rombach et al. (2022); Poole et al. (2023); Esser et al. (2024) have demonstrated impressive performance in real-world tasks such as high-quality image synthesis and image editing. However, such practical models usually require extensive computational resources to train, as well as a large number of model evaluations to generate high-quality samples (e.g. images). Using these pre-trained teacher models to generate pairs of random noises and their corresponding clean images, a popular choice for low-cost training and fast sampling is to train a student distillation model with advanced "trajectories" within the pair Salimans and Ho (2022); Wang et al. (2023, 2024); Yin et al. (2024b); Luo et al. (2024); Xie et al. (2024); Nguyen and Tran (2024); Yin et al. (2024a); Sauer et al. (2024); Xu et al. (2024); Zhu et al. (2025); Frans et al. (2025a); Sauer et al. (2025).

We can group these trajectory-based distillation methods into two categories: *consistency model distillation* Song et al. (2023); Kim et al. (2024); Lu and Song (2025) and *rectified flow distillation* Liu

---

[*]Corresponding author: `xuhui.fan@mq.edu.au`

39th Conference on Neural Information Processing Systems (NeurIPS 2025).

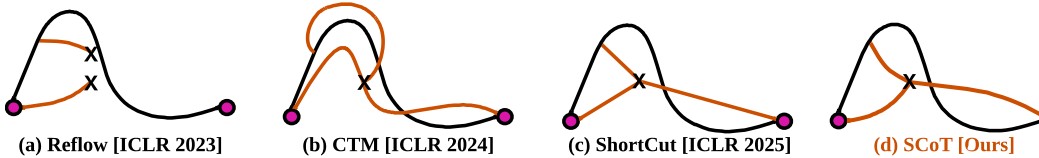

| (a) Reflow [ICLR 2023] | (b) CTM [ICLR 2024] | (c) ShortCut [ICLR 2025] | (d) SCoT [Ours] |

Figure 1: Comparison of trajectory distillation methods. The black line in each panel denotes the teacher trajectory of pre-trained diffusion models, which are connected within the pair of a random noise (left dot) and a clean image (right dot). The red solid line is the student trajectory of the distillation model. Panel (a) Reflow Liu et al. (2023b) straightens its student trajectory by enforcing its velocity be close to a constant. However, its trajectory maps different points to different values due to the lack of consistency. Panel (b) CTM Kim et al. (2024) places consistency requirement for the student trajectory. However, it might be difficult to track the student trajectory when it is of high curvatures. In panel (c), the Shortcut model Frans et al. (2025b) focuses on velocity estimation, while uses straight lines to approximate the trajectory and ensure the consistency. Our proposed SCoT model in panel (d) enforces straightness for consistent student trajectory. By avoiding the approximating errors of solving ODEs and by straightening the student trajectory, SCoT successfully bridges the gap between rectified flows and CTM distillations and enjoys the benefits of both approaches.

et al. (2023c); Zhu et al. (2025). Consistency model distillation focuses on the trajectory itself, requiring a "valid" consistent trajectory. That is, the student trajectory should map to the same point regardless of different initial points. Rectified flow distillation seeks a straight student trajectory by enforcing its velocities be close to the magnitude of point changes. While consistency distillation directly models pointwise changes along the trajectory, the straight trajectory of rectified flow distillation may reduce the number of steps required to generate high-quality images.

However, both categories have their own limitations. Consistency model distillation typically requires multiple steps to generate high-quality samples Song et al. (2023); Kim et al. (2024). This is likely due to the challenge of tracking the mapping function for trajectories with high curvatures. In the case of rectified flow distillation, numerical ODE solvers are required for a learned velocities, which introduce approximation errors that degrade the final output quality.

In order to address these issues and enjoy the benefits of both approaches, we bridge the gap between consistency models and rectified flows by proposing the **S**traight-**Co**nsistent **T**rajectories (SCoT) model in this paper. SCoT aims to produce trajectories that are consistent, which is a valid condition for trajectories, and are straight, which simplifies the point changes in the trajectory. In detail, SCoT regulates the trajectory through two aspects: (1) *trajectory straightness* by optimizing velocities towards the amount of point changes in the pre-trained model, (2) *trajectory consistency* by ensuring points from different time steps are projected to converge at the same value at future time steps.

In summary, the proposed SCoT is the first to produce consistent and straight trajectories between random noise and clean images, which unifies the consistency models and rectified flows. Consistency guarantees valid trajectories, while straightness facilitates the approximation of the trajectory projection function and faster sample generation. With this design, SCoT achieves state-of-the-art results, enabling compact models to generate high-quality data in $N$-steps, or even a single step. Figure 1 highlights the key differences among consistency distillation, rectified flow distillation, ShortCut model Frans et al. (2025b), and the proposed SCoT.

## 2 A trajectory perspective on Pre-trained diffusion model distillations

**Notation** We consider the trajectory to be working within the time interval $[0, 1]$, with the time steps specified as $1 = t_N > t_{N-1} > \ldots > t_1 = 0$. Since image synthesis is a common task for diffusion models, we denote $\mathbf{x}_0$ as the clean image, $\mathbf{x}_t$ as the noisy image at the time step $t$, and $\mathbf{x}_1 \sim \mathcal{N}(\mathbf{0}, \mathbf{I})$ as the random noise sampled from a standard Gaussian distribution. Furthermore, we use $\boldsymbol{\theta}$ and $\phi$ to represent the parameters of the pre-trained teacher model and the student distillation model, respectively. Detailed descriptions of the notation are provided in **??** and Table 7.

**Diffusion models** Diffusion models (DMs) Ho et al. (2020); Song et al. (2021a); Dhariwal and Nichol (2021) generate data by learning the score function of noisy images at multiple noise scales. At each

time step $t$, a *clean* image $\mathbf{x}_0$ is first diffused into a noisy image $\mathbf{x}_t$ through a forward process $\mathbf{x}_t := \alpha_t \mathbf{x}_0 + \sigma_t \mathbf{x}_1, \mathbf{x}_1 \sim \mathcal{N}(\mathbf{0}, \mathbf{I})$, where $\alpha_t$ and $\sigma_t^2$ are the diffusion coefficient and variance, respectively. Using $p_t(\mathbf{x}_t)$ to denote the probability of diffusion in $\mathbf{x}_t$, DMs learn a neural network $\boldsymbol{\epsilon}_{\boldsymbol{\theta}}(\mathbf{x}_t, t)$ that matches the score of the corrupted image $\mathbf{s}_{\text{real}}(\mathbf{x}_t) := \nabla_{\mathbf{x}_t} \log p_t(\mathbf{x}_t) = -\sigma_t^{-1}(\mathbf{x}_t - \alpha_t \mathbf{x}_0)$ by minimizing the loss $\mathbb{E}_{t, \mathbf{x}_t} \left[ \omega(t) \| \boldsymbol{\epsilon}_{\boldsymbol{\theta}}(\mathbf{x}_t, t) - \mathbf{s}_{\text{real}}(\mathbf{x}_t) \|^2 \right]$, where $\omega(t)$ is a weighting function.

Rectified flows Lipman et al. (2023); Liu et al. (2023b); Albergo et al. (2023) can be regarded as an extension of diffusion models by defining a linear interpolation between random noise $\mathbf{x}_1$ and a clean image $\mathbf{x}_0$ as $\mathbf{x}_t = t\mathbf{x}_1 + (1-t)\mathbf{x}_0, 0 \le t \le 1$. In details, they use the ODE $\mathrm{d}\mathbf{x}_t/\mathrm{d}t = \mathbf{v}_{\boldsymbol{\theta}}(\mathbf{x}_t, t)$ to transport between the noise distribution $\mathcal{N}(\mathbf{0}, \mathbf{I})$ and data distribution $\boldsymbol{\pi}_0(\mathbf{x}_0)$, whereas the velocity $\mathbf{v}_{\boldsymbol{\theta}}(\mathbf{x}_t, t)$ is learned by minimizing the loss $\mathbb{E}_t \left[ \| (\mathbf{x}_1 - \mathbf{x}_0) - \mathbf{v}_{\boldsymbol{\theta}}(\mathbf{x}_t, t) \|^2 \right]$.

Given a random noise $\mathbf{x}_1 \sim \mathcal{N}(\mathbf{0}, \mathbf{I})$, a new clean image $\widehat{\mathbf{x}}_0$ can be generated either through an iterative denoising process with a trained $\boldsymbol{\epsilon}_{\boldsymbol{\theta}}(\mathbf{x}_t, t)$ (in DMs), or through an ODE solver as $\widehat{\mathbf{x}}_0 = \mathbf{x}_1 + \int_1^0 \mathbf{v}_{\boldsymbol{\theta}}(\mathbf{x}_t, t)\mathrm{d}t$ (in rectified flows). In either case, a pair of random noise and its corresponding clean image is formed as $(\mathbf{x}_1, \widehat{\mathbf{x}}_0)$. Trajectory distillation focuses on the construction of advanced trajectories to accelerate sampling without compromising image qualities.

**Reflow distillation** The teacher trajectory produced by the pre-trained velocity $\mathbf{v}_{\boldsymbol{\theta}}(\mathbf{x}_t, t)$ may not be straight, since their training set $\mathbf{x}_1 \sim \mathcal{N}(\mathbf{0}, \mathbf{I})$ and $\mathbf{x}_0 \sim \boldsymbol{\pi}_0(\mathbf{x}_0)$ are not paired. To address this issue, reflow distillation Liu et al. (2023a,c); Zhu et al. (2025); Li et al. (2025) works on the pair $(\mathbf{x}_1, \widehat{\mathbf{x}}_0)$ by training a new velocity $\mathbf{v}_{\boldsymbol{\phi}}(\mathbf{x}_t, t)$ that approximates the direction $\widehat{\mathbf{x}}_0 - \mathbf{x}_1$ within the pair. With the trained $\mathbf{v}_{\boldsymbol{\phi}}(\mathbf{x}_t, t)$ expected to approximate the straight direction from $\mathbf{x}_1$ to $\widehat{\mathbf{x}}_0$, reflow distillations may use fewer steps to generate high-quality images. Other approaches uses one neural network to approximate the magnitude of changes over the whole time period Liu et al. (2023c).

**Consistency distillation** Consistency model (CM) Song et al. (2023) can be regarded as one trajectory distillation method. CM studies a consistent projection function $f_{\boldsymbol{\phi}}(\mathbf{x}_t, t)$ that maps any noisy image $\mathbf{x}_t$ to the clean image $\mathbf{x}_0$: $f_{\boldsymbol{\phi}}(\mathbf{x}_t, t) = \mathbf{x}_0, \forall t \in [0, 1]$. Its distillation objective function can be written as a weighted distance metric $\mathbf{D}(\cdot, \cdot)$ between the mapped clean images from two adjacent points $\mathcal{L}_{\text{CM}} = \mathbb{E}_t \left[ \omega(t) \mathbf{D} \left( \mathbf{f}_{\boldsymbol{\phi}}(\mathbf{x}_{t+\Delta t}, t + \Delta t), \mathbf{f}_{\boldsymbol{\phi}^-}(\widehat{\mathbf{x}}_t^{\boldsymbol{\theta}}, t) \right) \right]$, where $\phi^-$ is the exponential moving average of the past values $\phi$, and $\widehat{\mathbf{x}}_t^{\boldsymbol{\theta}}$ is obtained from the pre-trained model as $\widehat{\mathbf{x}}_t^{\boldsymbol{\theta}} = \mathbf{x}_{t+\Delta t} - t\Delta t \nabla_{\mathbf{x}_{t+\Delta t}} \log p_{t+\Delta t}(\mathbf{x}_{t+\Delta t})$. CM obtains a "valid" trajectory by mapping different points to the same clean image.

Building on CM, Consistency Trajectory Model (CTM) Kim et al. (2024) introduces a multi-step consistent mapping function defined as:

$$G_{\boldsymbol{\phi}}(\mathbf{x}_t, t, s) = (s/t)\mathbf{x}_1 + (1 - s/t)g_{\boldsymbol{\phi}}(\mathbf{x}_t, t, s), \tag{1}$$

in which $g_{\boldsymbol{\phi}}(\mathbf{x}_t, t, s)$ is left unconstrained, and is parameterized by a neural network $g_{\boldsymbol{\phi}}(\cdot)$. Equation (1) ensures $G_{\boldsymbol{\phi}}(\mathbf{x}_t, t, s)$ satisfies the boundary condition as $G_{\boldsymbol{\phi}}(\mathbf{x}_1, 1, 1) = \mathbf{x}_1$.

**Concurrent works** MeanFlow Geng et al. (2025) is a concurrent work to SCoT that also unifies consistency models and rectified flows via velocity integration. Its key distinction lies in introducing a correction term derived from the gradient with respect to the starting step $t$, unlike SCoT's use of the gradient at the terminating step $s$. Although rectified flow is regarded as a special case of MeanFlow under an infinitesimal integration range, it is not clear if MeanFlow itself can learn straight trajectories in practice. Another concurrent work FlowMap Sabour et al. (2025); Boffi et al. (2025) also shares the same target. By using the CTM as the backbone, it is unclear if such a setting can achieve both straight and consistent trajectories in practice.

# 3 Straight-Consistent Trajectories (SCoT) model

Given a pre-trained noise-image pair $(\mathbf{x}_1, \widehat{\mathbf{x}}_0)$, the proposed Straight-Consistent Trajectories (SCoT) model aims to learn a trajectory-based projection function $G_{\boldsymbol{\phi}}(\mathbf{x}_t, t, s)$ that produces straight and consistent trajectories within the pair. Similar to the CTM, the projection function $G_{\boldsymbol{\phi}}(\mathbf{x}_t, t, s)$ takes the current time step $t$, the values $\mathbf{x}_t$ at step $t$, as well as the future time step $s$ as inputs, and outputs the values at time step $s$ as $\widehat{\mathbf{x}}_s := G_{\boldsymbol{\phi}}(\mathbf{x}_t, t, s)$.

### 3.1 Main Components of SCoT

We introduce two regulators that refine the SCoT trajectory regarding its velocity and consistency.

**Constant-valued velocity** In order to ensure that SCoT produces a straight trajectory, we regularize the gradient of its projection function $G_\phi(\mathbf{x}_t, t, s)$. In detail, we encourage the partial derivative of $G_\phi(\mathbf{x}_t, t, s)$ with respect to $s$ to approximate the magnitude of the change observed within the pre-trained pair $(\mathbf{x}_1, \widehat{\mathbf{x}}_0)$. In this way, the velocity loss function can be defined as:

$$\mathcal{L}_{\text{velocity}} = \mathbb{E}_{\mathbf{x}_1, t, s}\left[\|\partial G_\phi(\mathbf{x}_t, t, s)/\partial s - (\widehat{\mathbf{x}}_0 - \mathbf{x}_1)\|^2\right], \tag{2}$$

where $t, s$ are sampled based on the step schedule.

Let $\boldsymbol{\mu}_\phi(\mathbf{x}_s, s)$ denote the velocity of the student trajectory at time step $s$ and let $\mathbf{x}_t$ denote the value at initial step $t$, $G_\phi(\mathbf{x}_t, t, s)$ is equivalent to the solution of the ODE $\mathrm{d}\widehat{\mathbf{x}}_s/\mathrm{d}s = \boldsymbol{\mu}_\phi(\widehat{\mathbf{x}}_s, s), \forall s \in [t, 1]$. That is:

$$\boxed{\widehat{\mathbf{x}}_s = G_\phi(\mathbf{x}_t, t, s) = \mathbf{x}_t + \int_t^s \boldsymbol{\mu}_\phi(\mathbf{x}_r, r)\mathrm{d}r \Rightarrow \frac{\partial G_\phi(\mathbf{x}_t, t, s)}{\partial s} = \boldsymbol{\mu}_\phi(\mathbf{x}_s, s)} \tag{3}$$

Equation (2) can be alternatively understood as enforcing the student trajectory's velocity $\boldsymbol{\mu}_\phi(\mathbf{x}_s, s)$ close to the magnitude of changes in the pre-trained teacher model.

In fact, this alternative formulation of the velocity loss $\|\boldsymbol{\mu}_\phi(\mathbf{x}_s, s) - (\widehat{\mathbf{x}}_0 - \mathbf{x}_1)\|^2$ is equivalent to that of the reflow distillation. By minimizing $\mathcal{L}_{\text{velocity}}$, SCoT achieves the same straightening effect on the student trajectory as reflow distillation. For reflow distillation, it learns the velocity of trajectory and thus needs to numerically solve ODEs to obtain the trajectory. SCoT directly learns the student trajectory through $G_\phi(\mathbf{x}_t, t, s)$ and avoids such ODE approximation errors. More importantly, SCoT can be regularized to satisfy the consistency requirement.

We compute the partial derivative $\partial G_\phi(\mathbf{x}_t, t, s)/\partial s$ using PyTorch's `torch.autograd.grad` function, which enables automatic differentiation of $G_\phi(\mathbf{x}_t, t, s)$ with respect to $s$.

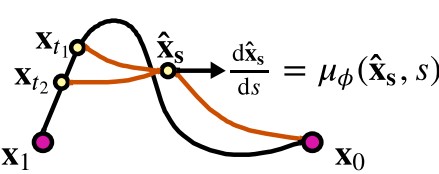

Figure 2: From two different points $\mathbf{x}_{t_1}, \mathbf{x}_{t_2}$, SCoT maps to the *same* point $\widehat{\mathbf{x}}_s$. The velocity $\boldsymbol{\mu}_\phi(\widehat{\mathbf{x}}_s, s)$ at time step $s$ is independent from previous time steps $t_1, t_2$.

**Trajectory consistency** SCoT also requires the trajectory to be consistent Song et al. (2023); Kim et al. (2024); Frans et al. (2025a). That is, the projection function $G_\phi(\mathbf{x}_t, t, s)$ maps the same future value to all points along the trajectory, regardless of their current time step as $G_\phi(\mathbf{x}_{t_1}, t_1, s) = G_\phi(\mathbf{x}_{t_2}, t_2, s)$, where $s < t_1 < t_2$. $\mathbf{x}_{t_1}$ is obtained through teacher model's ODE solver as $\mathbf{x}_{t_1} = \texttt{Solver}(\mathbf{x}_{t_2}, t_2, t_1; \boldsymbol{\theta}) = \mathbf{x}_{t_2} + \int_{t_2}^{t_1} \mathbf{v}_\theta(\mathbf{x}_r, r)\mathrm{d}r$, such that $\mathbf{x}_{t_1}$ and $\mathbf{x}_{t_2}$ are located on the same teacher trajectory.

We adopt the *soft-consistency loss* Kim et al. (2024) which is defined by comparing the mapped outputs $G_\phi(\mathbf{x}_{t_2}, t_2, s) \approx G_{\text{sg}(\phi)}(\texttt{Solver}(\mathbf{x}_{t_2}, t_2, t_1; \boldsymbol{\theta}), t_1, s)$, where $\text{sg}(\cdot)$ is the exponential moving average stop-gradient operator. Following CTM Kim et al. (2024), these two mapping values are further projected to the clean image space to construct the consistency loss as:

$$\mathcal{L}_{\text{consistency}} = \mathbb{E}_{t_2 \in [0,1], t_1 \in [t_2, 1], s \in [t_1, 1], \mathbf{x}_{t_2}}\left[D(\mathbf{x}_{\text{target}}(\mathbf{x}_{t_2}, t_2, t_1, s), \mathbf{x}_{\text{est}}(\mathbf{x}_{t_2}, t_2, s))\right], \tag{4}$$

where $D(\cdot, \cdot)$ is the distance metric, $\mathbf{x}_{\text{est}}(\mathbf{x}_{t_2}, t_2, s) = G_{\text{sg}(\phi)}(G_\phi(\mathbf{x}_{t_2}, t_2, s), s, 0)$ represents the estimated clean image obtained by projecting the point $\mathbf{x}_{t_2}$ forward to $s$ using the student model $G_\phi$, followed by decoding to time step 0 via the EMA model $G_{\text{sg}(\phi)}$, and $\mathbf{x}_{\text{target}}(\mathbf{x}_{t_2}, t_2, t_1, s) = G_{\text{sg}(\phi)}(G_{\text{sg}(\phi)}(\texttt{Solver}(\mathbf{x}_{t_2}, t_2, t_1; \boldsymbol{\theta}), t_1, s), s, 0)$ serves as the target output, computed by first integrating $\mathbf{x}_{t_2}$ to $\mathbf{x}_{t_1}$ along the teacher trajectory via the ODE solver, and then applying the same projection-decoding pipeline. This formulation ensures that both paths—although starting from different intermediate states on the same trajectory—should yield consistent outputs at the final clean image space.

This consistency ensures the "validity" of our projection function in the trajectory. Compared to the consistency models and their variants, it is expected that this trajectory straightness makes it easier to approximate the projection function than those with high curvatures.

**Objective function** Summarizing the above targets, SCoT's objective function may be written as:

$$\mathcal{L}_{\text{SCoT}} = \lambda_{\text{vel}}\mathcal{L}_{\text{velocity}} + \lambda_{\text{con}}\mathcal{L}_{\text{consistency}}, \tag{5}$$

where $\lambda_{\text{vel}}, \lambda_{\text{con}}$ denote the weighting factors for $\mathcal{L}_{\text{velocity}}, \mathcal{L}_{\text{consistency}}$. Each individual component addresses the straightness and consistency of trajectories, respectively. In this way, the proposed SCoT successfully unifies the consistency models and rectified flows.

The two loss terms, $\mathcal{L}_{\text{velocity}}$ and $\mathcal{L}_{\text{consistency}}$, have different dependencies on the pre-trained model. Specifically, $\mathcal{L}_{\text{velocity}}$ operates on a data-noise pair $(\widehat{\mathbf{x}}_0, \mathbf{x}_1)$, while $\mathcal{L}_{\text{consistency}}$ requires the model's parameters. To cope with their requirements, we propose generating the data point $\widehat{\mathbf{x}}_0$ dynamically for each training step using the pre-trained model $\widehat{\mathbf{x}}_0 = \mathbf{x}_1 + \int_1^0 \mathbf{v}_{\boldsymbol{\theta}}(\mathbf{x}_t, t)dt$. This avoids the need for pre-computed pairs. An additional potential improvement involves using a shared random noise vector $\mathbf{x}_1$ for both losses, which may enhance training consistency when sampling intermediate points. We leave a detailed study of this method to future work.

## 3.2 Training from scratch and Sampling

It is noted that SCoT may also be trained from scratch, by modifying consistency loss and randomly choosing noise $\mathbf{x}_1$ and data $\mathbf{x}_0$. Such exploration was not conducted due to resource constraints. Regarding sampling, Algorithm 1 outlines the SCoT sample generation process, which may be implemented as a multi-step or single step procedure. Table 2 and Table 3 demonstrate the corresponding experimental validation. Following the same setting as in Table 2 and Table 3, our model is trained with the DSM loss adopted from CTM to enhance sample quality and training stability.

## 3.3 Connections to Other Distillation Methods

| Methods | **Reflow** Liu et al. (2023b) | **InstaFlow** Liu et al. (2023c) |
|---|---|---|
| **Objective function** | $\mathbb{E}_{\mathbf{x}_1,t}\left[\|\mathbf{v}_{\boldsymbol{\phi}}(\mathbf{x}_t,t) - (\widehat{\mathbf{x}}_0 - \mathbf{x}_1))\|^2\right]$ | $\mathbb{E}_{\mathbf{x}_1}\left[\|\mathbf{v}_{\boldsymbol{\phi}}(\mathbf{x}_1,1) - (\widehat{\mathbf{x}}_0 - \mathbf{x}_1)\|^2\right]$ |
| **Methods** | **FlowDreamer** Li et al. (2025) | $\mathcal{L}_{\text{velocity}}$ in **SCoT** **(Ours)** |
| **Objective function** | $\mathbb{E}_{\mathbf{x}_t}\left[\|\mathbf{v}_{\boldsymbol{\phi}}(\mathbf{x}_t,t) - \mathbf{v}_{\boldsymbol{\theta}}(\mathbf{x}_t,t)\|^2\right]$ | $\mathbb{E}_{\mathbf{x}_1,t,s}\left[\|\partial G_{\boldsymbol{\phi}}(\mathbf{x}_t,t,s)/\partial s - (\widehat{\mathbf{x}}_0 - \mathbf{x}_1)\|^2\right]$ |

Table 1: Different objective functions in enforcing straight student trajectory. In addition to Reflow and SCoT, InstaFlow enables one step sampling by learning the magnitude of changes based on random noise $\mathbf{x}_1$, whereas FlowDreamer aims to approximate teacher model's velocity $\mathbf{v}_{\boldsymbol{\theta}}(\mathbf{x}_t, t)$.

The proposed SCoT bridges the gap between the consistency model and rectified flow distillations. On one hand, SCoT adopts the format of CTM Kim et al. (2024)'s project function $G_{\boldsymbol{\phi}}(\mathbf{x}_t, t, s)$. SCoT's consistency loss and output reconstruction loss can be combined into CTM's integrated *soft-consistency loss*. While there is no restriction on velocity, approximating $G_{\boldsymbol{\phi}}(\mathbf{x}_t, t, s)$ can be challenging when the student trajectory exhibits high curvature. As a result, multiple steps are usually required to generate high-quality images in CTM Kim et al. (2024).

---
**Algorithm 1** Sampling Procedure of SCoT

**Input:** Trained SCoT projection $G_{\boldsymbol{\phi}}(\mathbf{x}_t, t, s)$;
    steps $t_N = 1 > \cdots > t_1 = 0$
**Output:** Generated image $\widehat{\mathbf{x}}_0$
1: Sample initial: $\widehat{\mathbf{x}}_{t_N} \sim \pi_1$
2: **for** $n = N, N-1, \ldots, 1$ **do**
3:     $\widehat{\mathbf{x}}_{t_{n-1}} = G_{\boldsymbol{\phi}}(\widehat{\mathbf{x}}_{t_n}, t_n, t_{n-1})$
4: **end for**
5: **return**  $\widehat{\mathbf{x}}_0$

---

On the other hand, the velocity loss in SCoT shares the same target as Reflow Liu et al. (2023b). By enforcing constant velocity in Equation (2), SCoT induces a straightening effect on the student trajectory. Table 1 summarizes the different objective functions used to encourage straight student trajectories.

The Shortcut model Frans et al. (2025a) shares the same targets of trajectory consistency and straightness. While emphasizing learning trajectory velocities instead of the full trajectory, the Shortcut model still relies on an ODE solver for image sampling. Also, its usage of a one-step Euler solver to enforce trajectory consistency might not be optimal. Other concurrent works such

as MeanFlow Geng et al. (2025) and FlowMap Boffi et al. (2025); Sabour et al. (2025) have been discussed in Section 2.

Boot Gu et al. (2023) also considers the velocity alignment with the pre-trained model, by comparing the gradient of function. InstaFlow Liu et al. (2023c) and SlimFlow Zhu et al. (2025) focus on output reconstruction. As velocity and consistency are not regularized, the trajectory is not well defined.

# 4 Experiments

## 4.1 Experimental Setup

**Teacher-Student Distillation Setup.** To simulate the generation trajectory from $t = 0$ to $t = 1$, the pre-trained models are used to generate paired data samples $\langle \mathbf{x}_0, \mathbf{x}_1 \rangle$. Furthermore, to provide the intermediate states required for computing the consistency term in Equation (4), we distill knowledge from a teacher model $\theta$ into a student model $\phi$. Following established practices in CM Song et al. (2023), SlimFlow Zhu et al. (2025), and CTM Kim et al. (2024), we evaluate SCoT on both CIFAR-10 and ImageNet, using pre-trained diffusion checkpoints from EDM[2] (for CIFAR-10) and CM[3] (for ImageNet) as teacher models. For the student model, we adopt EDM's DDPM++ implementation on CIFAR-10, and the Ablated Diffusion Model (`ADM`) architecture from Dhariwal and Nichol (2021) for ImageNet. To support the additional time conditioning $s$ in Equation (5), we incorporate several architectural modifications inspired by CTM. Specifically, we extend the temporal embedding by adding auxiliary $s$-embedding information to the original $t$-embedding. We also adopt other training heuristics from CTM (see Table 8), including: (1) using a larger value of $\mu$ in the stop-gradient EMA to slow the teacher update rate and improve training stability; (2) setting the student EMA rate to 0.999 to smooth parameter updates over time and reduce training noise; and (3) reusing skip connections from the pre-trained diffusion model to facilitate gradient flow and preserve hierarchical features.

**Model Architectures.** To support the additional time conditioning variable $s$ in our generator $g_\phi(\mathbf{x}, t, s)$, we incorporate a conditional embedding module into the baseline model architectures, following prior design choices in CM Song et al. (2023), SlimFlow Zhu et al. (2025), and CTM Kim et al. (2024). For CIFAR-10, we adopt the DDPM++ implementation from EDM Karras et al. (2022), extending its time embedding module to additionally encode $s$ using a structure symmetric to the original $t$ embedding, and applying identical normalization strategies. For ImageNet, we use the Ablated Diffusion Model (`ADM`) Dhariwal and Nichol (2021), where $s$ is embedded jointly with the class conditioning variable $c$, requiring no architectural change. A detailed comparison of DDPM++ and ADM in terms of their ResNet backbones, attention configurations, and noise conditioning strategies is presented in Table 9. To improve the attention module, we address compatibility issues in `QKVFlashAttention` by modifying the dimension operations in `QKVAttentionLegacy` to match expected checkpoint formats. Additionally, we integrate `xformers'` `ScaledDotProduct` attention as an alternative backend. These improvements ensure correct weight loading while enhancing efficiency and flexibility of the attention mechanism.

**Training and Sampling Hyperparameters.** To ensure stable convergence and fair comparison during the training of SCoT, we adopt a higher learning rate for smaller datasets (e.g., CIFAR-10) and scale the batch size appropriately for larger datasets (e.g., ImageNet). We use the Adam optimizer for both settings, and enable mixed-precision training (FP16) to improve memory efficiency and computational speed. For CIFAR-10, we train the model for 130k iterations with a batch size of 512, while for ImageNet we use a larger batch size of 2048 and train for 40k iterations to accommodate the increased data complexity and training cost. We adopt an exponential moving average (EMA) of the student model with a decay rate of 0.9999 and employ a stop-gradient variant with EMA coefficient $\mu = 0.9999$ to stabilize the training objective. A complete list of hyperparameter settings is provided in Table 8.

---

[2] https://github.com/NVlabs/edm
[3] https://github.com/openai/consistency_models

| NFE↓ | Methods | Model Size | | | Generation Quality |
|---|---|---|---|---|---|
| | | FLOPs (G) | MACs (G) | Params (MB) | FID↓ |
| 50 | DDIM Song et al. (2021a) | 12.2 | 6.1 | 35.7 | 4.67 |
| 20 | DDIM Song et al. (2021a) | 12.2 | 6.1 | 35.7 | 6.84 |
| 10 | DDIM Song et al. (2021a) | 12.2 | 6.1 | 35.7 | 8.23 |
| 10 | DPM-solver-2 Lu et al. (2022) | 12.2 | 6.1 | 35.7 | 5.94 |
| 10 | DPM-solver-fast Lu et al. (2022) | 12.2 | 6.1 | 35.7 | 4.70 |
| 10 | 3-DEIS Zhang and Chen (2022) | 20.6 | 10.3 | 61.8 | 4.17 |
| 2 | PD Salimans and Ho (2022) | 41.2 | 20.6 | 55.7 | 5.58 |
| 2 | CD Song et al. (2023) | 41.2 | 20.6 | 55.7 | 2.93 |
| 2 | CT Song et al. (2023) | 41.2 | 20.6 | 55.7 | 5.83 |
| 2 | CTM* Kim et al. (2024) | 41.2 | 20.6 | 55.7 | 1.87 |
| 2 | **SCoT** | 41.2 | 20.6 | 55.7 | 2.30 |
| 1 | 1-Rectified Flow (+Distill) Liu et al. (2023a) | 20.6 | 10.3 | 61.8 | 378 |
| 1 | 2-Rectified Flow (+Distill) Liu et al. (2023a) | 20.6 | 10.3 | 61.8 | 12.21 |
| 1 | 3-Rectified Flow (+Distill) Liu et al. (2023a) | 20.6 | 10.3 | 61.8 | 8.15 |
| 1 | CTM* Kim et al. (2024) | 41.2 | 20.6 | 55.7 | 1.90 |
| 1 | **SCoT** | 41.2 | 20.6 | 55.7 | 2.40 |

Table 2: Comparison of $N$-step (NFE) generation performance across diffusion models on CIFAR-10 at comparable model scales. We report sample quality metrics—FID ($\downarrow$)-for $N \in \{1, 2, 10, 20, 50\}$. Entries highlighted in **bold** denote our proposed method. CTM* indicates the inclusion of GAN loss. Baseline results are sourced from Song et al. (2023); Zhu et al. (2025); Frans et al. (2025b). "–" indicates that the result is not available.

**Datasets.** For evaluation, we adopt two large-scale real-world datasets with different image resolutions: CIFAR-10 ($32 \times 32$) and ImageNet ($64 \times 64$), following standard protocol. Dataset statistics, including resolution, total size, and sample count, are summarized in Table 10.

**Evaluation Metrics.** We assess SCoT on unconditioned image generation using standard evaluation metrics, including Fréchet Inception Distance (FID) Heusel et al. (2017), Inception Score (IS) Salimans et al. (2016), and Recall (Rec.) Sajjadi et al. (2018). To evaluate model efficiency, we also report parameter count, floating-point operations (FLOPs), and multiply–accumulate operations (MACs). See Section C.2 for further details.

**Time Efficiency.** We measure the training throughput of the trajectory generator $g_\phi(\cdot)$ under different combinations of loss functions to evaluate time efficiency. Throughput results, reported in images per second per GPU, are summarized in Table 11.

## 4.2 Distillation Results

**CIFAR-10.** On CIFAR-10, SCoT achieves an FID of 2.30 with only 2 NFEs, outperforming baselines such as CD (FID 2.93) and CT (FID 5.83) while requiring fewer function evaluations. CTM attains a lower FID of 1.87 with just 2 NFE, but its results incorporate a GAN loss, which significantly improves sample fidelity by providing a strong adversarial signal during training. In contrast, we do not adopt the GAN loss due to its instability under our training configuration and resource constraints. Despite this, SCoT delivers competitive performance without adversarial training, demonstrating a favorable balance between generation quality and computational efficiency.

**ImageNet.** On ImageNet, SCoT achieves an FID of 2.60 with just 2 NFEs, outperforming baselines such as CD (FID 6.20) and PD (FID 15.39), and closely approaching the performance of EDM (FID 2.44). While CTM achieves the best FID of 1.92, this result benefits from the integration of a GAN loss, which enhances visual fidelity by introducing adversarial supervision during training. In contrast, our method does not incorporate GAN loss due to its instability under our training regime, yet still attains competitive sample quality. SCoT also demonstrates strong diversity, with a recall of 0.61 and an Inception Score of 68.20, highlighting its effectiveness in balancing quality, diversity, and computational cost.

| NFE↓ | Methods | Model Size | | | Generation Quality | | |
|---|---|---|---|---|---|---|---|
| | | FLOPs | MACs | Params | FID↓ | Rec.↑ | IS↑ |
| 250 | ADM Karras et al. (2022) | — | — | — | 2.07 | 0.63 | — |
| 79 | EDM Dhariwal and Nichol (2021) | 219.4 | 103.4 | 295.9 | 2.44 | 0.67 | 48.88 |
| 2 | PD Salimans and Ho (2022) | 219.4 | 103.4 | 295.9 | 15.39 | 0.62 | — |
| 2 | CD Song et al. (2023) | 219.4 | 103.4 | 295.9 | 6.20 | 0.63 | 40.08 |
| 2 | CTM* Kim et al. (2024) | 219.4 | 103.4 | 295.9 | 1.70 | 0.57 | 64.29 |
| 2 | **SCoT** | 219.4 | 103.4 | 295.9 | 2.60 | 0.61 | 68.20 |
| 1 | CD Song et al. (2023) | 219.4 | 103.4 | 295.9 | 6.20 | 0.63 | – |
| 1 | CT Song et al. (2023) | 219.4 | 103.4 | 295.9 | 13.00 | 0.47 | – |
| 1 | SlimFlow Zhu et al. (2025) | 67.8 | 31.0 | 80.7 | 12.34 | — | — |
| 1 | Shortcut(unconditional) Frans et al. (2025b) | 219.4 | 103.4 | 295.9 | 20.50 | — | — |
| 1 | Shortcut(conditional) Frans et al. (2025b) | 219.4 | 103.4 | 295.9 | 40.30 | — | — |
| 1 | CTM* Kim et al. (2024) | 219.4 | 103.4 | 295.9 | 1.92 | 0.57 | 64.29 |
| 1 | **SCoT** | 219.4 | 103.4 | 295.9 | 4.80 | 0.57 | 67.60 |

Table 3: Comparison of $N$-step generation performance by different DMs on ImageNet across corresponding model scales. Bold red numbers indicate the number of parameters for each distilled model. We report sample quality metrics—FID (↓), Rec. (↑), and IS (↑)—for $N \in \{1, 2, 10, 79, 250\}$. Entries highlighted in **bold** denote our proposed method. CTM* indicates the use of an additional GAN loss. Baseline results are taken from Song et al. (2023); Zhu et al. (2025); Frans et al. (2025b). "–" indicates that the result is not available.

## 4.3 Trajectory Analysis

We use CTM's soft consistency which proves to outperform local consistency and perform comparable to global consistency. Specifically, local consistency distills only 1-step teacher, so the teacher of time interval $[0, T - \Delta t]$ is not used to train the neural jump starting from $\mathbf{x}_T$. Rather, teacher on $[t - \Delta t, t]$ with $t \in [0, T - \Delta t]$ is distilled to student from neural jump starting from $\mathbf{x}_t$, not $\mathbf{x}_T$. The student, thus, has to extrapolate the learnt but scattered teacher across time intervals to estimate the jump from $\mathbf{x}_T$, which could potentially lead to imprecise estimation. In contrast, the amount of teacher to be distilled in soft consistency is determined by a random u, where $\mu = 0$ represents distilling teacher on the entire interval $[0, T]$. Hence, soft matching serves as a computationally efficient and high-performing loss.

## 4.4 Consistency Guarantee

The results in Table 4 highlight the benefits of consistency. By adding this loss, SCoT is encouraged to focus on improving output reconstruction and consistency, which is reflected in the consistent reduction of FID from 15.7 to 5.6 in NFE = 1 and from 16.4 to 3.9 in NFE = 2. This optimization drives the model to generate samples closer to the target distribution. As a result, the IS increases from 30.4 to 63.1 in NFE = 1 and from 29.8 to 61.4 in NFE = 2, indicating notable improvements in both fidelity and diversity. Additionally, Precision and Recall show substantial gains rising from 0.49 to 0.72 and 0.45 to 0.57 in NFE = 1, respectively, demonstrating that the model captures more accurate and diverse samples.

## 4.5 Velocity Guarantee

The incorporation of the velocity guarantee loss, as shown in Table 5, further improves the generative trajectory by encouraging temporal smoothness. This constraint ensures that the trajectories are straightened, making the sampling process more stable and efficient. The benefits are evident in the substantial reduction of FID, dropping from 14.7 to 4.8 in NFE = 1 and from 15.2 to 3.6 in NFE = 2. The IS improves accordingly, reaching 67.6 and 68.2 in NFE = 1 and NFE = 2, respectively. The model also achieves stronger coverage and accuracy, with Precision improving from 0.56 to 0.71 and Recall from 0.54 to 0.58 in NFE = 1. Similar trends are observed in NFE = 2. These confirm that velocity regularization helps refine the generative path, enabling the model to produce high-quality samples with better alignment to the underlying data manifold, especially under reduced NFE.

| Metric | NFE = 1 | | | | NFE = 2 | | | |
|---|---|---|---|---|---|---|---|---|
| | 5k | 10k | 15k | 20k | 5k | 10k | 15k | 20k |
| **FID** ($\downarrow$) | 15.7 | 15.2 | 11.4 | **5.6** | 16.4 | 14.7 | 10.2 | **3.9** |
| **sFID** ($\downarrow$) | 33.8 | 32.8 | 31.7 | **16.2** | 36.5 | 34.9 | 29.7 | **13.6** |
| **IS** ($\uparrow$) | 30.4 | 34.2 | 43.6 | **63.1** | 29.8 | 33.7 | 42.2 | **61.4** |
| **Precision** ($\uparrow$) | 0.49 | 0.55 | 0.64 | **0.72** | 0.47 | 0.53 | 0.65 | **0.73** |
| **Recall** ($\uparrow$) | 0.45 | 0.49 | **0.57** | 0.54 | 0.44 | 0.47 | **0.58** | 0.56 |

Table 4: Comparison of training results for NFE = 1 and NFE = 2 for loss from Equation (4). All models are trained on ImageNet and tested on 6k samples. Bold values indicate the best performance per metric.

| Metric | NFE = 1 | | | | NFE = 2 | | | |
|---|---|---|---|---|---|---|---|---|
| | 5k | 10k | 15k | 20k | 5k | 10k | 15k | 20k |
| **FID** ($\downarrow$) | 14.7 | 14.1 | 10.4 | **4.8** | 15.2 | 13.8 | 9.1 | **2.6** |
| **sFID** ($\downarrow$) | 32.0 | 31.0 | 29.9 | **14.6** | 35.4 | 33.7 | 28.3 | **12.4** |
| **IS** ($\uparrow$) | 33.4 | 37.8 | 48.6 | **67.6** | 34.1 | 38.5 | 50.0 | **68.2** |
| **Precision** ($\uparrow$) | 0.56 | 0.61 | 0.67 | **0.71** | 0.55 | 0.60 | 0.68 | **0.73** |
| **Recall** ($\uparrow$) | 0.54 | 0.56 | **0.58** | 0.57 | 0.55 | 0.57 | **0.59** | 0.58 |

Table 5: Training Equation (5) Results for NFE = 1 and NFE = 2. For metrics with $\downarrow$, lower values are better; for metrics with $\uparrow$, higher values are better. Bold values represent the best performance in each metric.

## 4.6 Loss Weighting

In this section, we explore the impact of loss weighting parameters in our training objective defined by Equation (5), focusing specifically on $\lambda_{con}$ for the consistency loss and $\lambda_{vel}$ for the velocity loss.

**Weighting on $\lambda_{con}$ in Equation (5)** We implemented an adaptive weighting strategy for $\lambda_{con}$ to effectively balance the contributions between consistency and denoising losses during training. The primary motivation behind this adaptive mechanism is to dynamically adjust the weighting based on the relative importance and magnitude of each loss component. Specifically, this adjustment is made through gradient magnitude comparisons, ensuring that no single loss dominates excessively, thereby stabilizing training and improving convergence.

**Weighting on $\lambda_{vel}$ in Equation (5)** To address the instability associated with the velocity loss defined in Equation (2), we explored several loss weighting strategies. The primary source of instability stems from the computation of second-order derivatives using automatic differentiation tools such as PyTorch's `torch.autograd.grad()`. These second-order computations require repeated differentiation, which can amplify small numerical inaccuracies through successive applications of the chain rule. As a result, accumulated numerical errors may lead to unstable gradient magnitudes, manifesting as gradient explosion or vanishing gradients, and ultimately impairing training stability. Motivated by these challenges, we investigated alternative weighting approaches to mitigate such instabilities.

The first strategy, **Adaptive Weighting**, dynamically adjusts the weight of the velocity loss based on the scale of individual loss elements. As shown in Table 6, this method performs relatively well in the early stages of training, but FID increases notably in later iterations. This indicates that while adaptive scaling can help stabilize early training, it may overemphasize certain loss components later on, leading to degraded generative quality.

The second strategy, **Without Scaling**, removes all adaptive scaling factors. This approach is motivated by the observation that unstable second-order gradients may produce extreme weight values, which can disrupt training. As evidenced in Table 6, this method leads to slightly worse performance at early iterations but achieves improved stability and better FID scores in later stages.

The final strategy, **Normalized Velocity Loss**, clips the adaptive weight within a predefined range $[0.01, 10]$ to prevent extreme values from dom-

|  | 5k | 10k | 15k | 20k |
|---|---|---|---|---|
| **Adaptive Weighting** | 15.3 | 14.5 | 11.2 | 6.2 |
| **Without Scaling** | 16.1 | 15.4 | 12.5 | 7.0 |
| **Normalization** | 15.2 | 13.8 | 9.1 | 3.6 |

Table 6: FID ($\downarrow$) comparison across different velocity loss weighting strategies at various training iterations (5k, 10k, 15k, 20k), using a global batch size of 2048. **Adaptive Weighting** dynamically adjusts weights based on the relative scale of loss components; **Without Scaling** removes adaptive weighting entirely; and **Normalization** stabilizes training by bounding the magnitude of the velocity loss.

inating the loss. This normalization consistently achieves the best FID scores across all iterations, suggesting it provides a stable and balanced contribution of the velocity loss during optimization. Notably, after 20k iterations, this strategy yields the lowest FID, demonstrating its effectiveness in maintaining training stability and improving overall model performance.

## 5 Conclusion

In this paper, we propose the SCoT model that successfully bridges the gap between consistency model and rectified flows distillation. SCoT performs pre-trained diffusion model distillations, and produces straight and consistent trajectories for fast sampling. By aligning the output, velocity, and consistency with the teacher model, SCoT achieves high-quality generation with fewer sampling steps. Experimental results on CIFAR-10 and ImageNet show that SCoT outperforms existing distillation methods in both efficiency and generation quality. Ablation studies confirm the effectiveness of our design, highlighting the importance of trajectory consistency and velocity approximation. In the future, we aim to extend SCoT to high-resolution image synthesis and explore its integration with conditional generation tasks. We believe SCoT provides a strong foundation for efficient generative modeling and paves the way for real-time high-fidelity image synthesis.

## 6 Acknowledgments and Disclosure of Funding

This work is sponsored by AMD's High Performance Compute Fund project. This work is also partially sponsored by the Australian Research Council Discovery grant DP240102050, ARC LIEF grant LE240100131, and ARC Linkage grant LP230201022.

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

# Contents

## A Theoretical Motivation for Loss Design

To better understand the design of our loss function, we provide a brief explanation of why combining velocity and consistency objectives helps the model perform better.

The trajectory projection function is defined as

$$\hat{\mathbf{x}}_s = G_\phi(\mathbf{x}_t, t, s) = \frac{s}{t}\mathbf{x}_t + \left(1 - \frac{s}{t}\right) g_\phi(\mathbf{x}_t, t, s), \tag{6}$$

where $g_\phi$ is a neural network that learns a residual correction. The first term encourages a straight path from $\mathbf{x}_t$ to the target, while the second term provides flexibility.

The velocity loss encourages the trajectory to follow a constant direction:

$$\mathcal{L}_{\text{velocity}} = \mathbb{E}_{x_1,t,s}\left[\left\|\frac{\partial G_\phi}{\partial s} - (\hat{\mathbf{x}}_0 - \mathbf{x}_1)\right\|^2\right]. \tag{7}$$

The consistency loss encourages points from different timesteps to align at the same target point:

$$\mathcal{L}_{\text{consistency}} = \mathbb{E}_{t_1,t_2,s}\left[\|G_\phi(\mathbf{x}_{t_1}, t_1, s) - G_\phi(\mathbf{x}_{t_2}, t_2, s)\|^2\right]. \tag{8}$$

These two losses serve different purposes. The velocity loss reduces trajectory curvature, making it easier to approximate with fewer steps. The consistency loss ensures that the mapping is valid across time. Together, they help improve both the quality and efficiency of generation, especially in low-step settings like 1-step or 2-step sampling.

## B Related Work

Pre-trained diffusion distillation methods have emerged as a powerful strategy to alleviate the significant computational burden inherent in diffusion models, which traditionally require a large number of function evaluations to produce high-quality samples. By compressing the iterative denoising process into far fewer steps, these techniques not only expedite sample generation but also make it feasible to deploy such models in low-resource environments. In this work, we systematically categorize these distillation approaches into three major groups based on their underlying objectives and operational paradigms.

**Output reconstruction based methods:** These aim to minimize the discrepancy between the outputs of the teacher (i.e., the pre-trained diffusion model) and the student model by directly reconstructing image outputs. Some approaches, such as Progressive Distillation Salimans and Ho (2022), focus on aligning output values by enforcing a close correspondence between the denoising steps of the teacher and the student. Other methods, for instance SDS and its variants Poole et al. (2023); Wang et al. (2024); Yin et al. (2024b,a), concentrate on matching output distributions to preserve the statistical characteristics of generated images. Additionally, certain studies operate in a one-step denoising image space Lukoianov et al. (2024); Karras et al. (2022), allowing for the direct generation of high-quality images from a single function evaluation, while others employ Fisher divergence objectives Zhou et al. (2024, 2025) to more rigorously align the gradients of score functions. Together, these techniques effectively reduce the number of sampling steps required while maintaining the fidelity of generated outputs.

**Trajectory distillation based methods:** Instead of concentrating solely on the final output, trajectory-based methods focus on the entire denoising path—from the initial random noise to the eventual clean image. By distilling the full trajectory, these approaches ensure that the student model replicates not only the final result but also the dynamic behavior of the teacher model throughout the diffusion process. Consistency distillation techniques Song et al. (2023); Kim et al. (2024); Lu and Song (2025) emphasize the self-consistency of the denoising trajectory, ensuring stable and accurate progression across different time steps. In contrast, rectified flow distillation methods such as InstaFlow and SlimFlow Liu et al. (2023c); Zhu et al. (2025) focus on producing a straighter, more direct trajectory, thereby mitigating the accumulation of approximation errors that typically arise from curved paths. Moreover, recent studies have demonstrated that integrating consistency modeling directly into rectified flows Frans et al. (2025a) can further enhance the fidelity of generated trajectories, effectively combining the strengths of both approaches.

**Adversarial distillation based methods:** This category of distillation methods leverages adversarial learning to refine the student model's output distribution. By incorporating an adversarial loss—often implemented via a pre-trained classifier or discriminator, these methods drive the student model to more closely approximate the target distribution provided by the teacher. Notably, studies such as Sauer et al. (2024, 2025) have successfully employed this strategy to achieve competitive performance with significantly fewer sampling steps. The adversarial framework not only enhances the perceptual quality of generated images but also provides a flexible plug-and-play mechanism that can complement other distillation strategies.

| Notation | Description |
|---|---|
| $\phi$ | Parameters of the generator/student model |
| $\theta$ | Parameters of the velocity/teacher model |
| $t$ | Current time step ($t \in [0, 1]$) |
| $s$ | Target time step for state projection |
| $N$ | Number of discretized steps/evaluations |
| $\pi_0$ | Data distribution at initial state ($\mathbf{t} = 0$) |
| $\pi_1$ | Noise distribution at terminal state ($\mathbf{t} = 1$) |
| $g(\cdot)$ | Integration approximation function |
| $\mathbf{v}(\cdot)$ | Velocity prediction function |
| $G(\cdot)$ | Distilled generator function |

Table 7: Notations and corresponding descriptions used in training SCoT.

|  | Hyperparameters | CIFAR-10 | ImageNet |
|---|---|---|---|
| | Learning rate | 0.0004 | 0.000008 |
| | Batch | 512 | 2048 |
| | Student's stop-grad EMA parameter $\mu$ | 0.9999 | 0.9999 |
| | Optimizer | Adam | Adam |
| **Optimization** | N | 18 | 40 |
| | Training iterations | 130K | 40K |
| | Mixed-Precision (FP16) | Enabled | Enabled |
| | ODE Solver | Heun | Heun |
| **Score** | EMA decay rate | 0.999 | 0.999 |

Table 8: Training configuration of SCoT in different model sizes on CIFAR-10 and ImageNet.

## C   Technical Details

### C.1   Datasets

**CIFAR-10** [4]: This dataset contains 60,000 32×32 color images evenly distributed over 10 distinct classes (6,000 images per class). It is split into a training set of 50,000 images and a test set of 10,000 images, making it a standard benchmark in machine learning and computer vision.

**ImageNet** [5]: ImageNet is one of the most influential benchmarks in computer vision. It comprises over 1.2 million training images and around 50,000 validation images, categorized into 1,000 diverse classes. The dataset's vast scale and rich annotations have made it an essential resource for developing and evaluating deep learning models.

### C.2   Metrics

**Fréchet Inception Distance (FID).** FID evaluates the similarity between real and generated samples by modeling their feature distributions as multivariate Gaussians in a pre-trained InceptionV3 feature

---

[4]`https://www.cs.toronto.edu/~kriz/cifar.html`
[5]`http://www.image-net.org/`

| Component | Functionality | DDPM++ (CIFAR-10) | ADM (ImageNet) |
|---|---|---|---|
| ResNet Block | Downsampling method | Resize + Conv | Strided Conv |
| | Upsampling method | Resize + Conv | Transposed Conv |
| | Time embedding type | Fourier | Positional |
| | Normalization type | GroupNorm | LayerNorm |
| | Residual blocks per resolution | 2 | 3 |
| Attention Module | Applied resolutions | 16 | 32, 16, 8 |
| | Attention heads | 1 | 1–8–12 |
| | Attention blocks (Down) | 2 | 9 |
| | Attention blocks (Up) | 1 | 13 |
| | Attention implementation | Vanilla | Multi-head self-attention |
| Conditioning | Label embedding | None | Learned class embedding |
| | Extra temporal input ($s$) | Additive embedding | Additive embedding |
| | Positional encoding added | No | Yes |
| | Skip connections | Present | Present |
| Output Head | Output scaling | Yes | Yes |
| | Activation | Identity | Identity |

Table 9: Comparison of U-Net architectural components used in DDPM++ (CIFAR-10) and ADM (ImageNet) within our distillation framework. Differences include normalization strategies, attention configuration, conditioning, and up/downsampling methods.

| Data Shape | Dataset | Samples | Data Size |
|---|---|---|---|
| $(3 \times 32 \times 32)$ | CIFAR-10 | 60K | 160M |
| $(3 \times 64 \times 64)$ | FFHQ-64 | 70K | 5 GB |
| | ImageNet | 1.2M | 100 GB |

Table 10: Experimental details of datasets.

space and computing the Fréchet distance. This metric jointly captures fidelity and diversity. We adopt `clean-fid` [6] for FID computation, which standardizes image preprocessing and Inception activations, thereby improving reproducibility across experiments.

**Sliced Fréchet Inception Distance (sFID).** sFID is a computationally efficient variant of FID that approximates the Fréchet distance using one-dimensional projections of feature embeddings. Instead of computing the full covariance matrix, sFID calculates the Wasserstein distance between sliced marginal distributions in the InceptionV3 feature space, offering a lightweight and scalable measure of generation quality. We follow the same evaluation protocol as `clean-fid` [7], ensuring consistency in feature extraction and preprocessing.

**Precision and Recall**[8]**.** We adopt the manifold-based metrics introduced in Kynkäänniemi et al. (2019) to assess sample fidelity and diversity. Precision quantifies the fraction of generated samples that fall within the manifold of real data, while Recall measures how much of the real data distribution is covered by generated samples. The manifolds are approximated via $k$-nearest neighbors in the InceptionV3 feature space. Our implementation follows the version in the ADM repository [9].

---

[6] `https://github.com/GaParmar/clean-fid`
[7] `https://github.com/GaParmar/clean-fid`
[8] `https://github.com/kynkaat/improved-precision-and-recall-metric`
[9] `https://github.com/openai/guided-diffusion`

**Inception Score (IS)**[10]. IS reflects both sample quality and class diversity by computing the KL divergence between the conditional class distribution and the marginal class distribution over all samples. It is calculated using logits from an InceptionV3 model Szegedy et al. (2016) trained on ImageNet Russakovsky et al. (2015). Higher scores suggest high-confidence predictions and class diversity. However, when evaluating datasets with limited categorical variation (e.g., CelebA or FFHQ), IS primarily reflects fidelity. We follow the ADM implementation [3] and evaluate IS using 10k generated images.

**Parameter Counts** (`MB`). We report the total number of learnable parameters in the model, measured in megabytes. This includes weights and biases from all trainable layers, such as convolutional filters and dense layer matrices. Higher parameter counts may indicate stronger representational capacity, while smaller models are better suited for deployment under resource constraints. For transformer-based architectures, we include attention and feed-forward components.

**Floating Point Operations (FLOPs).** FLOPs measure the total number of floating-point operations (additions and multiplications) required during a forward pass, serving as a proxy for computational cost. For a convolutional layer of kernel size $k \times k$ over a feature map of spatial size $H \times W$, the FLOPs are computed as:

$$\text{FLOPs} = 2 \times H \times W \times k^2 \times C_{\text{in}} \times C_{\text{out}}.$$

We compute FLOPs using the `calflops` utility [11], providing an estimate of overall model complexity.

**Multiply–Accumulate Operations (MACs).** MACs count the number of fused multiply-add computations, which are often optimized as single hardware instructions on modern accelerators. For the same convolutional layer, MACs are given by:

$$\text{MACs} = H \times W \times k^2 \times C_{\text{in}} \times C_{\text{out}}.$$

We report MACs using the same tool [2], as they offer a hardware-aware indicator of inference cost, especially relevant for edge deployment.

# D    Additional Experiments

## D.1    Time Efficiency

Table 11 concludes the throughput in training different loss function of our SCoT.

| Dataset | Equation (2) | Equation (4) |
|---------|--------------|--------------|
| CIFAR-10 | 282.3 | 71.5 |
| ImageNet | 131.7 | 31.7 |

Table 11: Time efficiency comparison (imgs/sec. on H100 GPU) of SCoT under different loss functions across datasets.

## D.2    Sample Comparisons

Figure 3, Figure 4, and Figure 5 present the qualitative results of 1-step image generation on ImageNet using the SCoT sampler described in Algorithm 1. These samples are taken at different stages of training to illustrate the progressive refinement of the model. Figure 3 shows samples generated at the initial stage of training, where the model produces blurry images with limited semantic structure. As training progresses to 10k steps (Figure 4), the generated images become more coherent, with clearer object boundaries and texture. At 30k steps (Figure 5), the model generates high-quality, semantically consistent images, demonstrating the effectiveness of the training process. To ensure efficient evaluation and maintain consistent comparison across stages, we generate 6,000 samples at each step. Following the same setting as in Table 2 and Table 3, our model is trained with the DSM loss adopted from CTM Kim et al. (2024) to enhance sample quality and training stability.

---

[10] https://github.com/openai/improved-gan

[11] https://github.com/MrYxJ/calculate-flops.pytorch

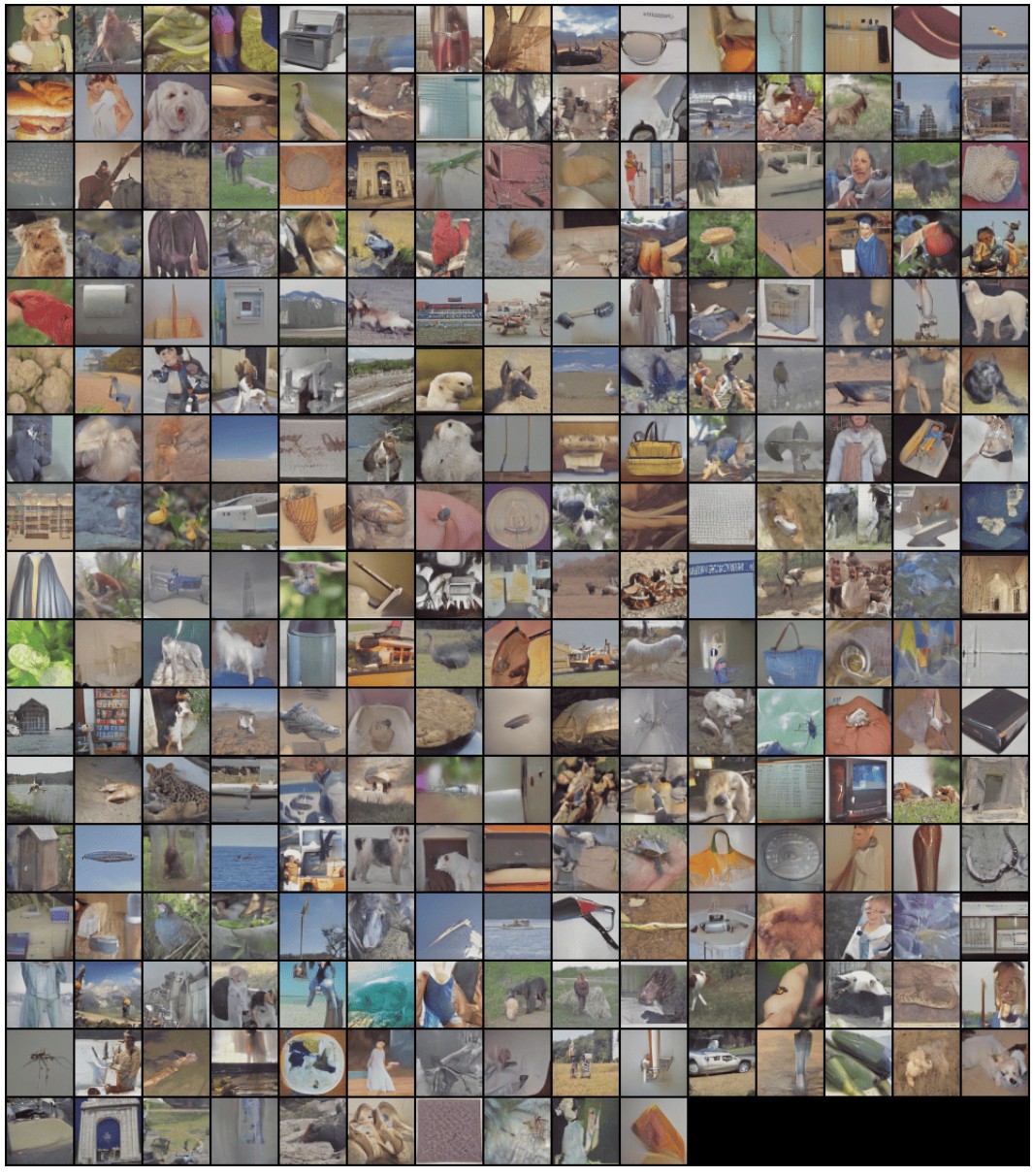

Figure 3: 1-Step generation on the initial training stage for ImageNet by SCoT Algorithm 1 sampler.

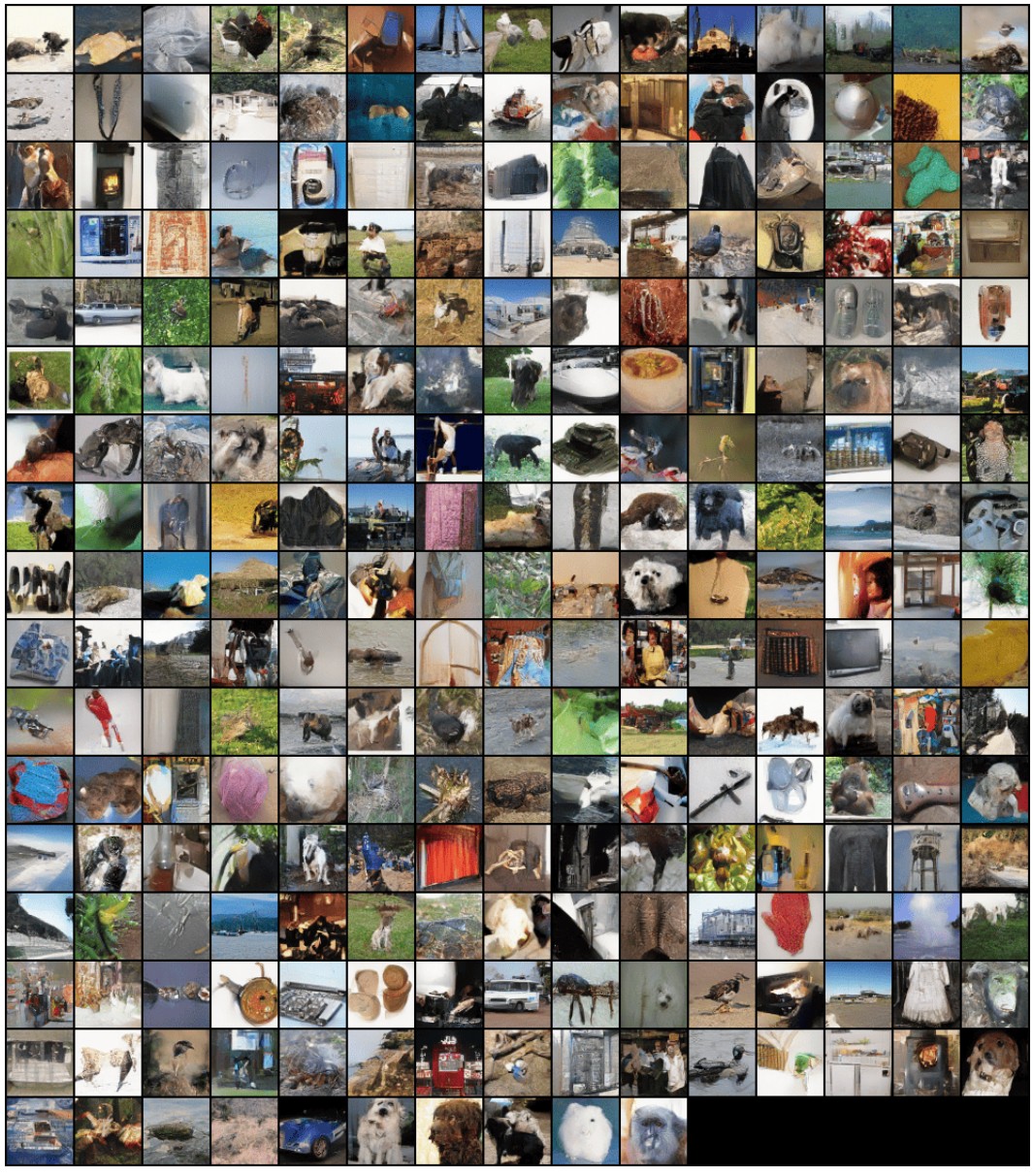

Figure 4: 1-Step generation on the 10k training stage for ImageNet by SCoT Algorithm 1 sampler.

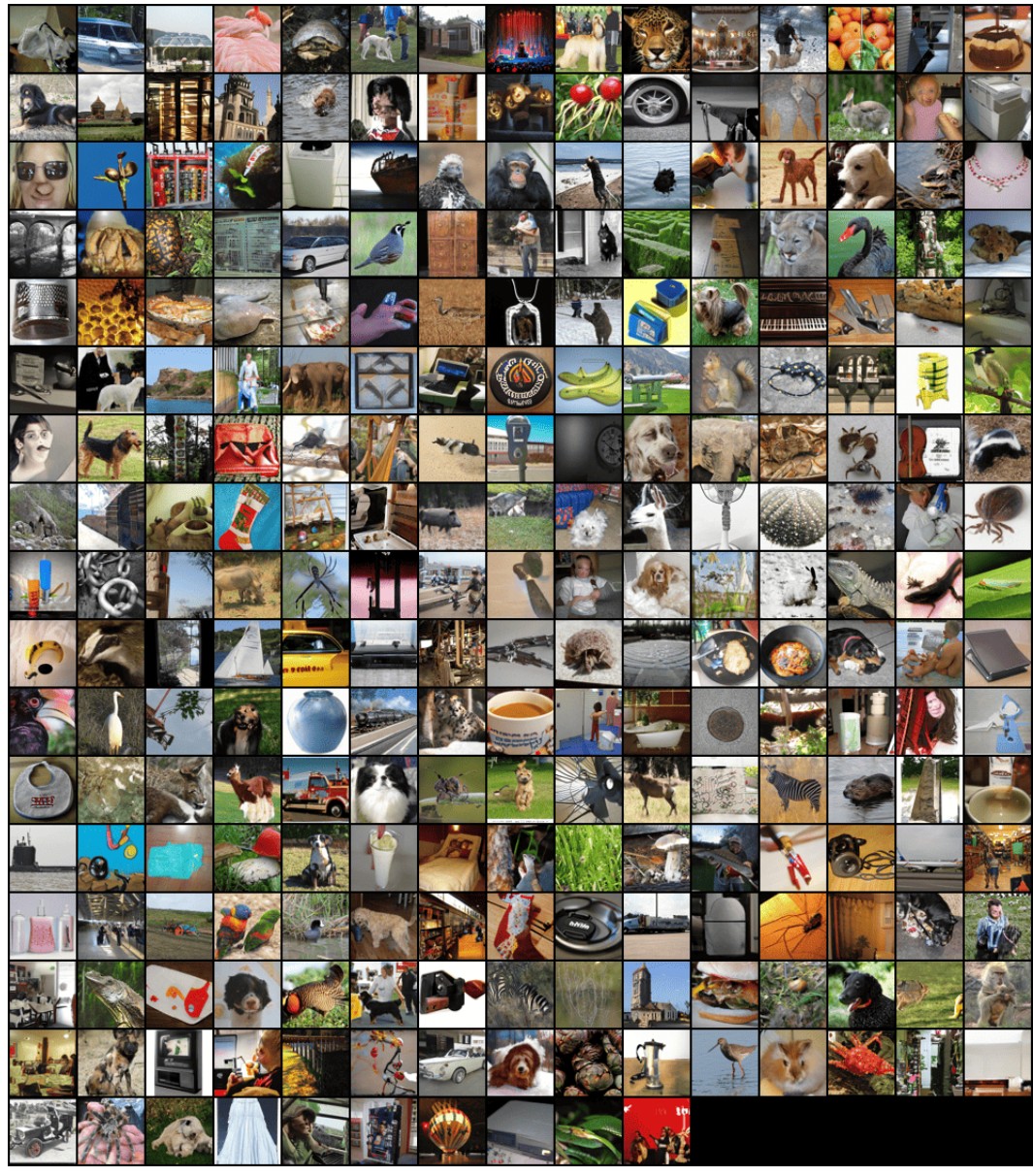

Figure 5: 1-Step generation on the 30k training stage for ImageNet by SCoT Algorithm 1 sampler.

