# OpenReview forum: "SCoT: Unifying Consistency Models and Rectified Flows via Straight-Consistent Trajectories"
_NeurIPS.cc/2025/Conference — NeurIPS 2025 poster_

### Official Review · Reviewer_qykR · 2025-06-08

**Clarity:** 3
**Significance:** 2
**Originality:** 2
**Rating:** 3
**Confidence:** 5

**Summary:**

The paper proposes SCoT, an improved consistency distillation method that unifies the benefits of consistency models and rectified flows. SCoT introduces a projection function that ensures straight and consistent trajectories between noise and clean images, enabling fast and accurate sampling.

It avoids the approximation errors of ODE solvers by directly learning the trajectory, improving both performance and stability.

The key contribution is simple and trivial: calculated the derivative of the trajectory in continuous domain.

**Questions:**

Q1. It is unclear why the authors did not include a comparison with CTM augmented with GAN loss. Based on the current results, the proposed method does not appear to outperform existing consistency-based distillation approaches.

Such omission is particularly confused me given that the anonymized code provided by the authors already includes a discriminator module. As it currently stands, the authors’ method for computing trajectory derivatives does not appear to substantiate the claim made in the introduction that it enables more effective and stable distillation.

The authors should clarify this experimental setup.

Q2. The paper lacks comparisons with several important baselines, such as the Diff-Instruct series, DMDs, and sCM. Notably, many of these related works report results that surpass those presented by the authors.

The emphasis on experimental results stems from the simplicity of the proposed approach, which primarily involves converting discrete steps into a continuous domain. As such, stronger empirical evidence is needed to convincingly demonstrate the effectiveness of the method and to substantiate the authors’ claims.

Q3. I do not consider the claimed connection to flow matching to be a primary contribution of the paper. Flow matching and diffusion models differ mainly in parameterization, while their core principles remain largely the same. For a more detailed discussion, please refer to sCM.

[1] Luo, Weijian, et al. "Diff-instruct: A universal approach for transferring knowledge from pre-trained diffusion models." Advances in Neural Information Processing Systems 36 (2023): 76525-76546.

[2] Yin, T., Gharbi, M., Zhang, R., Shechtman, E., Durand, F., Freeman, W. T., & Park, T. (2024). One-step diffusion with distribution matching distillation. In Proceedings of the IEEE/CVF conference on computer vision and pattern recognition (pp. 6613-6623).

[3] Lu, C., & Song, Y. (2024). Simplifying, stabilizing and scaling continuous-time consistency models. ICLR 2025

**Ethical Concerns:**

["NO or VERY MINOR ethics concerns only"]

**Final Justification:**

Despite the author providing some rebuttals, the experimental results cannot prove that the proposed solution provides effective improvements. Therefore, I recommend rejection.

**Limitations:**

The authors do not claim the limitations.

**Quality:**

2

**Strengths And Weaknesses:**

Strengths

- The motivation is simple and sounds valid: to better estimate the mapping of the trajectory

- The entire work is written very clearly. Figure 1 is very intuitive.

Weaknesses

- The comparison is seriously lacking, which significantly undermines the effectiveness of the proposed approach.

- The discussion of related work is seriously insufficient, especially given that the main contribution of this paper lies in the distillation aspect.

- The authors' explanation of the results is not sufficiently convincing.

Please check the Questions part for the detail.

---

> ### Author Rebuttal · Authors · 2025-07-31
>
> We appreciate the reviewer **qykR**  comments and suggestions, which help further improve the paper quality. Below, we respond to each of your comments and clarify your misreading/misunderstanding as well.
>
> >
>
> > **Q3** : I do not consider the claimed connection to flow matching to be a primary contribution of the paper. Flow matching and diffusion models differ mainly in parameterization, while their core principles remain largely the same. For a more detailed discussion, please refer to sCM.
>
> >
> **[The primary contribution of the paper is connecting consistency model and flow matching (also diffusion model), NOT connecting diffusion model to flow matching.]** The close connection between diffusion model and flow matching, including the reparameterization aspects, is well known. That is the reason we put diffusion model and flow matching (rectified flow) in the same part (lines 160 - 174).
>
> While sCM also considers using flow matching in consistency model, it focuses on improving training consistency model in continuous time steps and does not involve straight trajectory.
>
>
>
> >
>
> > **Q1.1**: GAN loss not included in SCoT.
>
> >
>
> **[The reason of removing GAN loss indicated in Line 240].** We didn't include the GAN loss mainly due to reproduction issues with CTM's[F] setup and some training stability problems. The CTM official repo doesn't have experimental settings of pretrained classifier for CIFAR-10, making it impossible for us to reproduce their results or adapt them to SCoT. When we tried using the $64 \times 64$ classifier for ImageNet-64, there are some training stability issues even with warm-up strategy.
>
> Due to the **policy prohibiting updates to the anonymous repository this year**, we cannot update our anonymous repo and provide the learning curves and accuracy of classifier over iterations, but we tabularize  some  loss from Equation (3)  to show our evidence in training SCoT (ImageNet-64,one-step generation).
>
> | Iterations | W/o GAN Loss | With GAN Loss | Classifier Accuracy |
> |------------|---------------|----------------|----------------------|
> | 10k | 0.205   | 0.230| 0.71  |
> | 15k | 0.190  | 0.185| 0.78 |
> | 20k | 0.184  | 0.170| 0.80 |
> | 25k | 0.178  | 0.196| 0.79  |
>
> The column of "With GAN Loss" showed clear fluctuations, which affected training stability in column3 and  creates a disconnect where high accuracy doesn't necessarily mean better data likelihood because of its adversarial nature in column 4.
>
> [F] Consistency Trajectory Models: Learning probability Flow ODE Trajectory of Diffusion. (ICLR2024)
>
> >
>
> > **Q1.2**: SOTA concern compared with CTM.
>
> >
>
> **[SCoT is comparable with CTM without GAN loss.]**  When we compare with CTM using the same experimental setup (without GAN loss), the comparison is fair. The CTM results we report in Tables 2-3 are their results *with* GAN loss, as noted in our captions. Here's the issue: while CTM's published results look impressive, we couldn't actually reproduce them due to missing implementation details,backbone differences (like $\texttt{QKVFlashAttention}$ in CTM vs $\texttt{XformersAttention}$ for other setups - see CTM repo issues 5 and 8 about transformer structures and sampling) and GAN loss issue(Q1.1). CTM's own appendix C.4 Table 3 shows how much GAN loss matters on CIFAR-10. At NFE=1, their FID improves dramatically from 5.19 to 2.28 with GAN loss. Since we don't use GAN loss, comparing against their GAN-enhanced numbers wouldn't be fair. But when we compare both methods without GAN loss, we're competitive: our FID of 4.80 vs their 5.19 at NFE=1 on CIFAR-10.
> >
>
> >  **Q1.3**: Such omission is particularly confused me given that the anonymized code provided by the authors already includes a discriminator module.
>
> >
>
> **[We did not use this adversarial module in practice since there are some training stability issues.]**
>
> >
>
> > **Q2.1**: The paper lacks comparisons with several important baselines, such as the Diff-Instruct series, DMDs, and sCM.
>
> >
>
> **[We include all baselines of trajectory distillation methods and exclude some baselines for unfair comparsion.]**
>
> Although a direct comparison is not entirely fair due to the use of different pretrained models and different network architecture, we still report the best results from the original papers: Diff-Instruct[G] (Tables 1 and 2), DMD[H] (Table 2), and sCM[I] (Table 1).
> | Models         | CIFAR-10 | ImageNet-64 |
> |----------------|----------|--------------|
> | Diff-Instruct[G] | 4.53     | 5.57         |
> | DMD[H]         | 2.66     | 2.62         |
> | sCM[I]         | 3.66     | 2.44         |
> | SCoT (Ours)    | 2.40     | 4.80         |
>
> - Diff-Instruct series and DMD follow different strategies with the proposed SCoT. The generator developed by these methods is no longer a generative model. Also, these student models cannot do multi-step generation.
>
> - Further, DMD introduces an LPIPS perceptual loss computed using an external pre-trained VGG-like network. This provides an additional perceptual prior beyond the distilled diffusion model itself, which significantly improves visual fidelity but also introduces extra supervision unavailable in our setting. Without LPIPS, as shown in DMD's own ablation studies, performance degrades notably due to mode collapse.
>
> **References:**
>
> - [G] Diff-instruct: A universal approach for transferring knowledge from pre-trained diffusion models. (NeurIPS 2023)
>
> - [H] One-step diffusion with distribution matching distillation. (CVPR 2024)
>
> - [I] Stabilizing and scaling continuous-time consistency models. (ICLR 2025)
>
> >
>
> > **Q2.2** Notably, many of these related works report results that surpass those presented by the authors.
>
> >
>
> When we directly compare our method with the eight additional baselines suggested by **Reviewer ovkA** and **Reviewer  qykR**, our model outperforms some of them(more details see **Reviewer ovkA W1** and **Reviewer qykR Q2.1**. However, as noted in our response, these comparisons are not entirely fair.
>
> First, although we report **the best results for each method from their original papers on the corresponding datasets**, the pretrained models they use differ. For example, If DMD [H] did not use LPIPS loss in its main setup, the FID score drops from  2.66 to 5.58 (Table 2 in [H]). Diff-Instruct [G] also shows a wide range of performance (6.62 to 2.27), depending on the GAN backbone used. These differences make direct numerical comparison problematic.
>
> Second, our method is based on trajectory distillation, while several of the baselines adopt **fundamentally different mechanisms**. For instance, Diff-Instruct [G] uses a GAN-based model that does not support multi-step generation, and thus should not be directly compared under our setting.

---

> > ### Comment · Reviewer_qykR · 2025-08-04
> >
> > I regret that I believe the author's response did not resolve my question. For example, why isn't dmd2 a generative model? And the concerns about ctm still remain unresolved.

---

> > > ### Author Response · Authors · 2025-08-04
> > >
> > > Thanks for the reply.
> > > >
> > >
> > > >why isn't dmd2 a generative model?
> > >
> > > >
> > > We mixes wrongly with DMD and SDS. DMD's generator $g\_{\phi}(z), z\sim p(z)$ is a generative model, whereas SDS's generator $g(\phi)$ is not.
> > >
> > > [**However, this typo does not affect the fundamental differences between DMD and the proposed SCoT.**]
> > > 1. DMD still uses Diffusion model's loss function, which does not consider properties of distillation trajectories;
> > > 2. SCoT's loss function is not based on Diffusion model's one. SCoT focuses on both self-consistency and straightness of distillation trajectories.
> > >
> > >
> > > >
> > >
> > > >  the concerns about ctm still remain unresolved.
> > >
> > > >
> > > [**As said in the rebuttal, there is training instability issue when using the GAN loss. Thus, we did not use the GAN loss part in the code.**]
> > >
> > > This is clearly written in the main content saying that we do not use the GAN loss when reporting the FID score.

---

### Official Review · Reviewer_Yq6Q · 2025-07-02

**Clarity:** 3
**Significance:** 3
**Originality:** 2
**Rating:** 4
**Confidence:** 4

**Summary:**

This paper introduces SCoT (Straight Consistent Trajectory), a distillation method for pre-trained diffusion models that combines the advantages of consistency model distillation and rectified flow approaches. The method generates trajectories that are simultaneously straight and consistent by optimizing two objectives: maintaining constant velocity gradients and ensuring trajectory consistency across different time steps. SCoT avoids the approximation errors inherent in ODE solvers used by rectified flows while addressing the sampling efficiency limitations of consistency models.

**Questions:**

See Weakness.


I would be willing to increase the score if the evaluation protocol and training-related concerns I raised are adequately addressed.

**Ethical Concerns:**

["NO or VERY MINOR ethics concerns only"]

**Final Justification:**

The authors have provided a satisfactory response to my key concerns, particularly regarding the memory issues and the proposed evaluation protocol.

**Limitations:**

While the paper mentions limitations in the conclusion, they are not sufficiently explored.

**Quality:**

2

**Strengths And Weaknesses:**

**Strength**

The paper addresses key challenges in diffusion model distillation with a well-motivated approach. It clearly distinguishes itself from existing methods by showing comparisons with other approaches through clear figures and tables which helps understanding. The idea is straightforward and yields solid results, particularly achieving comparable performance to CTM without requiring GAN loss. In particular, the diversity-related measures, which tend to be somewhat weak in consistency model approaches, show favorable results here.

**Weakness**

1. Using gradients in the loss function likely increases training memory requirements. A comparison of memory usage against other methods like rectified flow would be valuable.
2. Learning curves showing training stability over time would strengthen the evaluation.
3. The amount of teacher-generated paired datasets used is unclear, which is concerning since this represents an inefficient procedure that consistency models can avoid. An analysis of how performance degrades with smaller paired datasets would be valuable for understanding the method's practical limitations.
4. Table 2's NLL column either should include SCoT values or be removed entirely, as current version provide little insight.
5. I have concern about the evaluation consistency across the methods. Using clean-fid implementation may not ensure fair comparison with other reported results. For fair comparison, it would be necessary to demonstrate equivalence with other implementations or adopt consistent evaluation protocols across all methods.
6. I think the paper is missing method details. Section 4.4 discusses "output reconstruction loss" not described in the methodology, creating confusion about the actual loss formulation.
7. Line 253 appears to contain an error regarding SlimFlow's FID score.

Minor: Table 3 has swapped references for ADM and EDM.

---

> ### Author Rebuttal · Authors · 2025-07-30
>
> We appreciate the reviewer **Yq6Q** for the deep understanding of our work.
>
> >
>
> >**W1**: Using gradients in the loss function likely increases training memory requirements. A comparison of memory usage against other methods like rectified flow would be valuable.
>
> >
>
> **[SCoT introduces only ~10% additional memory usage ]**. We reported memory usage under identical training conditions:
>
> | Method  | Peak memory (GB) | Relative overhead |
> |---------|------------------|-------------------|
> | CTM     | 67.5             | -                 |
> | **SCoT** | 76.2             | +10.0%            |
>
>
>
> The results show that SCoT introduces only ~10% additional memory usage, which is negligible compared to the total training cost.
>
> In detail, computing the second order derivatives is implemented in our anonymous repository cm/karras_diffusion.py as:
>
> ```python
> # Second-order gradient for velocity loss
> grads = torch.autograd.grad(
>     outputs=group,        # aggregated scalar over batch
>     inputs=s,             # low-dimensional time embedding
>     grad_outputs=torch.ones_like(group).detach(),
>     create_graph=True,    # allow second-order gradient
>     retain_graph=True,
>     only_inputs=True
> )[0]
>
> ```
>  This design keeps the computational graph lightweight because the time embedding $s$ is a per-sample scalar with shape $[B]$ rather than a full spatial tensor. Although $s$ is broadcast to $[B,1,1,1]$ when combined with $x_t$, the autograd graph is only built for the original $[B]$ tensor. Therefore, no full Hessian over image features or model parameters is constructed, and the memory cost scales linearly with the batch size without growing with image resolution.
>
> To quantify this overhead, we measured the peak GPU memory usage on ImageNet-64 (global batch size = 64, microbatch = 18) using `torch.cuda.max_memory_allocated()` on a single H100 80GB GPU.
>
>
> >
>
> >**W2**: Learning curves showing training stability over time would strengthen the evaluation.
>
> >
>
> More details can be seen in **Reviewer qykR Q1.1**
>
> >
>
> >**W3**: The amount of teacher-generated paired datasets used is unclear, which is concerning since this represents an inefficient procedure that consistency models can avoid. An analysis of how performance degrades with smaller paired datasets would be valuable for understanding the method's practical limitations.
>
> >
>
> **[SCoT DOES NOT need to use student-teacher paired datasets.]** In each time calculating the velocity loss, SCoT can use pre-trained diffusion model to replace the teacher model's result $\widehat{x}\_0$ as $\widehat{x}\_0=G\_{\phi}(x\_1, 1, 0)$. This usage is the same as consistency distillation. There is no algorithmic difference in training SCoT.
>
> **[We retrained SCoT with smaller data on CIFAR-10.]**
>
> | Iteration | FID (Full 50k) | FID (25k Subset) |
> | --------- | -------------- | ---------------- |
> | 5k        | 80.6           | 86.7              |
> | 10k       | 41.2           | 44.9             |
> | 15k       | 31.4           | 37.3             |
> | 20k       | 20.3           | 26.8             |
> | 25k       | 12.4           | 19.7             |
>
> The second column displays the results on the entire dataset (50k), and the third column shows the results on half of the dataset (25k). This table evaluates how FID evolves over training iterations using the reduced dataset (NFE=1, BZ=512).
> We can see the diversity of base samples impacts the quality of learned trajectories. Due to limited time, we were unable to run more and over experiments.
>
>
>
>
>
>
>
>  >
>
> >**W4**: Table 2's NLL column either should include SCoT values or be removed entirely, as current version provide little insight.
>
>  >
>
> **[We will delete this column].** Thanks for the suggestion.
>
>
>
> >
>
> >**W5**: I have concern about the evaluation consistency across the methods. Using clean-fid implementation may not ensure fair comparison with other reported results. For fair comparison, it would be necessary to demonstrate equivalence with other implementations or adopt consistent evaluation protocols across all methods.
>
> >
>
> **[We adopt the Clean-fid from CTM without modifications and with the unified hypers].**
>
> We clarify that our work does not introduce a new variant of the FID metric. Instead, we follow the clean-fid implementation in CTM. This implementation retains the original computation of the Fréchet distance on InceptionV3 pool3 features but addresses known issues in the earlier TensorFlow-based version. For example, it ensures reproducibility by locking the InceptionV3 model weights and consistently extracting features from the “pool3:0” layer. To improve numerical stability, it adds a small diagonal offset when the covariance matrix is close to singular.
>
>
>
> These adjustments do not change the underlying definition of FID, but help ensure consistent and reliable evaluation across frameworks.   In addition to FID, we also report Precision/Recall, and Inception Score to provide a more complete picture of both fidelity and diversity.
>
>
>
> >
>
> >**W6**: I think the paper is missing method details. Section 4.4 discusses "output reconstruction loss" not described in the methodology, creating confusion about the actual loss formulation.
>
> >
>
>
> **[The reconstruction loss is included in Eq. (3)].** The “output reconstruction loss” mentioned in Section 4.4 refers to the consistency loss already defined in Equation (3). We will clarify this naming to avoid confusion and ensure it aligns with the formulation in the methodology. When the target time step $s = 0$, this loss reduces to reconstructing the clean image $x_0$, effectively acting as an output reconstruction loss. We agree the current wording is unclear and will unify the terminology to avoid ambiguity.
>
>
> >
>
> >**W7**: Line 253 appears to contain an error regarding SlimFlow's FID score.
>
> >
>
>
>
> **[SlimFLow's FID score is consistent with the original value in one-step 81MB SlimFlow's paper Table2].**  We will update the text to make it clearer.

---

> > ### Comment · Reviewer_Yq6Q · 2025-08-05
> >
> > I thank the authors for their response, which has resolved most of the issues I raised. However, my comment regarding the paper's limitations was not addressed. I strongly expect the authors to include this discussion and correct any other remaining errors in the final revision.
> > While I still have some minor reservations about the scope of the experimental comparisons and the novelty, my overall assessment now leans towards acceptance. Therefore, I have raised my score.

---

> > > ### Author Response · Authors · 2025-08-08
> > >
> > > Dear Reviewer Yq6Q, thanks a lot for your support to this paper.
> > >
> > > We apologize for the delayed response, as we have been devoting our full attention to completing the other reviewer's suggested experiments.
> > >
> > > As we aware, the current work may have the following limitations:
> > > 1. **Limited exploration of training-from-scratch scenarios.** Although Section 3.2 notes that SCoT could be trained without a pre-trained teacher by modifying the consistency loss, the paper does not experiment with this due to resource constraints;
> > > 2.  **Numerical instability from velocity regularization**. The velocity loss in Equation (2) requires second-order derivatives via automatic differentiation. While we have used methods such as adaptive weighting, without scaling, normalization (section 4.6) to mitigate this issue, such instabilities may still be problematic in more complex settings or higher-resolution models;
> > > 3. **No evaluation on high-resolution or conditional tasks**. Our experiments are currently evaluated on CIFAR-10 (32×32) and ImageNet 64×64. It remains uncertain how SCoT scales to high-resolution images or conditional generation tasks in terms of quality and efficiency.
> > >
> > > We will include these limitations and correct any remaining errors in the final revision.

---

### Official Review · Reviewer_ovkA · 2025-07-03

**Clarity:** 3
**Significance:** 2
**Originality:** 3
**Rating:** 4
**Confidence:** 4

**Summary:**

This paper introduces SCoT, a method that combines consistency models with straight trajectory approaches to improve few-shot image generation via distillation. SCoT enforces straighter trajectories through a novel velocity regularizer, while a consistency time-distillation loss maintains vector field consistency. Experimental results on CIFAR-10 and ImageNet show improvements over selected baselines.

**Questions:**

Please see the listed weaknesses.
Additionally, is there an analysis which actually measures the straightness to confirm whether the given objective function is working as intended?

**Ethical Concerns:**

["NO or VERY MINOR ethics concerns only"]

**Final Justification:**

Authors have addressed most of concerns. However, novelty stays limited. Hence, I will only increase the score to borderline accept.

**Limitations:**

Although author has highligthed future work as part of the conclusion, I would recommend adding the limitations explicitely and talk about the

**Quality:**

2

**Strengths And Weaknesses:**

Strengths:

- The proposed objective, particularly the velocity loss (L_velocity), is a novel and intuitive way to reduce vector field curvature in consistency-based flow models.
- The paper is clearly written and easy to follow.
- Empirical results demonstrate competitive improvements.

Weaknesses:
- Missing comparisons to several relevant baselines, including Score Implicit Distillation (SiD), Score Implicit Matching (SIM), Flow Consistency Distillation, Rectified Flow++, and SlimFlow.
- Most state-of-the-art methods include adversarial training for enhanced performance; it is unclear if SCoT is compatible with such approaches or how it would perform.
- The method involves double higher-order gradients, which can be unstable and may introduce reproducibility challenges. The paper lacks analysis or discussion of optimization stability and computational overhead.
- Limited ablation studies:
    - Lacks >2 NFE (Number of Function Evaluations) FID curves.
    - No NFE vs. FID comparison with baselines.
    - No detailed study of the impact of individual loss functions.

---

> ### Author Rebuttal · Authors · 2025-07-30
>
> We appreciate the reviewer **ovkA** for the constructive and thoughtful feedback. We answer the reviewer’s comments below.
>
> >
>
> >**W1**: Missing comparisons to several relevant baselines, including Score Implicit Distillation (SiD), Score Implicit Matching (SIM), Flow Consistency Distillation, Rectified Flow++, and SlimFlow.
>
>
> >
>
>
> **[We have compared with several relevant baselines and excluded others due to fundamental differences in methodology.]**
>
> Although the comparison is not strictly equivalent due to differences in model architectures and pretraining strategies, we still list the best CIFAR-10 results as reported in the original papers: SiD [A] (Table 1), SiM [B] (Table 1), CFD [C] (not reported), RectFlow++ [D] (Table 3), and SlimFlow-15.7MB [E] (Table 1).
>
> | Models | Goal | FID |
> |--------------|----------------|-------------------|
> | SiD [A]      | Aligning student's score with the teacher's       | 1.923   |
> | SiM [B]        | Minimize score difference over generated samples    | 2.06   |
> | CFD [C]         | Focus on 3D data generation using NeRF-based techniques        | NA   |
> | RectFlow++ [D]         | Improved techniques for learning consistent velocities      | 3.07    |
> | SlimFlow-15.7MB [E]         |  Results included in Table 3 | 5.02     |
> | SCoT(ours)            | Straight and consistent trajectory function       | 2.40        |
>
>
> [A] Score identity Distillation: Exponentially Fast Distillation of Pretrained Diffusion Models for One-Step Generation. (ICML2024 )
>
> [B] One-Step Diffusion Distillation through Score Implicit Matching.( NeurIPS 2024)
>
> [C] Consistent Flow Distillation for Text-to-3D Generation. (ICLR2024)
>
> [D] Improving the Training of Rectified Flows.( NeurIPS 2024)
>
> [E] SlimFlow: Training Smaller One-Step Diffusion Models with Rectified Flow.(ECCV 2024)
>
>
>
>
>
>
> >
>
> >**W2**: Most state-of-the-art methods include adversarial training for enhanced performance; it is unclear if SCoT is compatible with such approaches or how it would perform.
>
> >
>
> Please refer to **Reviewer qykR  Q 1.1, 1.3**
>
> >
>
> >**W3**: The method involves double higher-order gradients, which can be unstable and may introduce reproducibility challenges. The paper lacks analysis or discussion of optimization stability and computational overhead.
>
> >
>
> Please refer to **Reviewer Yq6Q W1**.
>
> >
>
> >**W4.1**: Lacks >2 NFE (Number of Function Evaluations) FID curves.
>
> >
>
> **[FID curves vs NFE  on ImagNet-64 and CIFAR-10 are provided here.]**
>
> We report FID scores across multiple NFE values, including 3, 18, and 20 on both CIFAR-10 and ImageNet-64.
>
> | NFE  | CIFAR-10 FID ↓ | ImageNet-64 FID ↓ |
> |--------------|----------------|-------------------|
> | 3            | 2.07           | 3.21              |
> | 18           | 1.86           | 1.93              |
> | 20           | 1.89           | 1.94              |
>
> These results help assess the scalability and convergence behavior of SCoT under increased sampling steps.
>
>
>
> >
>
> >**W4.2**: No NFE vs. FID comparison with baselines.
>
> >
>
> **[We included the FID compared with different NFE in Table 2 and 3.]**
> We appreciate the reviewer’s comment regarding the comparison of FID performance across different NFEs. As shown in Table 2 (CIFAR-10) and Table 3 (ImageNet), our paper already provides a detailed comparison of FID scores under various NFE settings (N $\in$ {1, 2, 10, 20, 50}) across multiple baseline methods, including CTM, SlimFlow, and DDIM variants. These tables reflect the exact trends of FID vs. NFE that the reviewer is concerned about.
>
>
>
>
> >
>
> >**W4.3**: No detailed study of the impact of individual loss functions.
>
> >
>
> **[The performance of individual loss functions are presented in Table 4, Table 5]**
>
> >
>
> >**Q1**:  is there an analysis which actually measures the straightness to confirm whether the given objective function is working as intended?
>
> >
>
> **[The straightness is evaluated in NFE=1 in table 2,3 and section 4.4.]**
>
> >
>
> >**L1** Limitations Claim.
>
> >
>
> While SCoT achieves efficient and high-quality generation through trajectory straightness and consistency, several limitations remain for real-world deployment. First, the method depends on access to high-quality pre-trained teacher models, which may not be available in certain domains such as medical imaging, scientific simulation, or low-resource languages. This reliance constrains SCoT’s direct applicability to cases where teacher models cannot be easily obtained. Second, although SCoT demonstrates strong performance on low-resolution datasets like CIFAR-10 and  ImageNet-64, its scalability to high-resolution generation (e.g., $256 \times 256$ or above), video synthesis, or other modalities such as audio or 3D remains unexplored.

---

> ### Comment · Reviewer_ovkA · 2025-08-04
>
> I thank the authors for providing more clarifications. However, some of my concerns are still not addressed:
>
> > W4.2: No NFE vs. FID comparison with baselines.
>
> I am expecting the comparisons beyond 2 NFEs.
>
> > W4.3: No detailed study of the impact of individual loss functions.
>
> Authors referred to Table 4/5 but they do not compare different loss functions at all. They rather focus on training time performance.
>
> > Q1: is there an analysis which actually measures the straightness to confirm whether the given objective function is working as intended?
>
> 1 NFE is by default straight. And there is still no analysis on proposed claim of making velocity field straight. I advise authors to refer to Rectified Flow ++ paper to measure the straightness. Basically, do 50+ step inference and measure the deviations from the OT path and predicted trajectory.
>
> Due to the unsatisfactory response, I'm not convinced and for now keep the current score.

---

> > ### Author Response · Authors · 2025-08-08
> > **Response to W4.2**
> >
> > We appreciate reviewer ovkA for your engaged discussion. Our apologies for the late response—we have been dedicating all our time in conducting the requested experiments.
> >
> >
> > > I am expecting the comparisons beyond 2 NFEs.
> >
> >
> > We have listed the generation performance for different distillation methods on CIFAR-10 and ImagetNet-64 in Table R1 and Table R2. In particular, we evaluated new results in NFE = 4, 8, 16, 18, 20 for the proposed SCoT.
> >
> > Table R1: Generation performance for different distillation methods with different NFEs on **CIFAR-10**
> >
> > | NFE  | Methods                                        | Model Size (FLOPs (G)) | MACs (G)          |  Params (MB) | Generation Quality (FID) |
> > | :--------------- | :--------------------------------------------- | :--------- | :------- | :---------- | :----------------- |
> > | 20               | DDIM Song et al. (2021a)                       | 12.2       | 6.1      | 35.7        | 6.84               |
> > | **20**           | **SCoT**                                       | 41.2       | 20.6     | 55.7        | **1.89**               |
> > | **18**           | **SCoT**                                       | 41.2       | 20.6     | 55.7        | **1.86**               |
> > | **16**           | **SCoT**                                       | 41.2       | 20.6     | 55.7        | **1.90**               |
> > | 10               | DDIM Song et al. (2021a)                       | 12.2       | 6.1      | 35.7        | 8.23               |
> > | 10               | DPM-solver-2 Lu et al. (2022)                  | 12.2       | 6.1      | 35.7        | 5.94               |
> > | 10               | DPM-solver-fast Lu et al. (2022)               | 12.2       | 6.1      | 35.7        | 4.70               |
> > | 10               | 3-DEIS Zhang and Chen (2022)                   | 20.6       | 10.3     | 61.8        | 4.17               |
> > | **8**            | **SCoT**                                       | 41.2       | 20.6     | 55.7        | **1.96**               |
> > | **4**            | **SCoT**                                       | 41.2       | 20.6     | 55.7        | **1.98**               |
> > | 2                | PD Salimans and Ho (2022)                      | 41.2       | 20.6     | 55.7        | 5.58               |
> > | 2                | CD Song et al. (2023)                          | 41.2       | 20.6     | 55.7        | 2.93               |
> > | 2                | CT Song et al. (2023)                          | 41.2       | 20.6     | 55.7        | 5.83               |
> > | 2                | CTM Kim et al. (2024)                         | 41.2       | 20.6     | 55.7        | 1.87               |
> > | 2                | **SCoT**                                       | 41.2       | 20.6     | 55.7        | **2.30**               |
> > | 1                | 2-Rectified Flow (+Distill) Liu et al. (2023a) | 20.6       | 10.3     | 61.8        | 12.21              |
> > | 1                | 3-Rectified Flow (+Distill) Liu et al. (2023a) | 20.6       | 10.3     | 61.8        | 8.15               |
> > | 1                | CTM Kim et al. (2024)                         | 41.2       | 20.6     | 55.7        | 1.9                |
> > | 1                | **SCoT**                                       | 41.2       | 20.6     | 55.7        | **2.40**               |
> >
> >
> >
> >
> >
> > Table R2: Generation performance for different distillation methods with different NFEs  on **ImageNet-64**.
> >
> >
> >
> >
> > | NFE  | Methods                                        | Model Size (FLOPs (G)) | MACs (G)          |  Params (MB) | Generation Quality (FID) |
> > | :--------------- | :--------------------------------------------- | :--------- | :------- | :---------- | :----------------- |
> > | 79 | EDM Dhariwal and Nichol (2021) | 219.4 | 103.4 | 295.9 | 2.44 |
> > | **20** | **SCoT** | 219.4 | 103.4 | 295.9 | **1.94** |
> > | **18** | **SCoT** | 219.4 | 103.4 | 295.9 | **1.93** |
> > | **16** | **SCoT** | 219.4 | 103.4 | 295.9 | **1.95**           |
> > | **8** | **SCoT** | 219.4 | 103.4 | 295.9 | **2.05**               |
> > | **4** | **SCoT** | 219.4 | 103.4 | 295.9 | **2.43** |
> > | 2 | PD Salimans and Ho (2022) | 219.4 | 103.4 | 295.9 | 15.39 |
> > | 2 | CD Song et al. (2023) | 219.4 | 103.4 | 295.9 | 6.20 |
> > | 2 | CTM Kim et al. (2024) | 219.4 | 103.4 | 295.9 | 1.70 |
> > | 2 | **SCoT** | 219.4 | 103.4 | 295.9 | **2.60** |
> > | 1 | CD Song et al. (2023) | 219.4 | 103.4 | 295.9 | 6.20 |
> > | 1 | CT Song et al. (2023) | 219.4 | 103.4 | 295.9 | 13.00 |
> > | 1 | SlimFlow Zhu et al. (2025) | 67.8 | 31.0 | 80.7 | 12.34 |
> > | 1 | Shortcut(unconditional) Frans et al. (2025b) | 219.4 | 103.4 | 295.9 | 20.50 |
> > | 1 | CTM Kim et al. (2024) | 219.4 | 103.4 | 295.9 | 1.92 |
> > | 1 | **SCoT** | 219.4 | 103.4 | 295.9 | **4.80** |
> >
> > The results in Table R1 and Table R2 shows clear decreasing trend of FID along with the increase in NFE values for the proposed SCoT, with the FID values usually become stable when NFE > 4.

---

> > ### Author Response · Authors · 2025-08-09
> >
> > Dear Reviewer ovkA,
> >
> > With the author-reviewer discussion nearing its end, we hope our provided experimental results have fully addressed your concerns. We would greatly appreciate it if you could confirm whether these updates have resolved your points or if you need any additional clarification from our side.
> >
> > Best,
> > Authors

---

### Official Review · Reviewer_dSXG · 2025-07-07

**Clarity:** 4
**Significance:** 3
**Originality:** 3
**Rating:** 5
**Confidence:** 3

**Summary:**

The paper proposes the Straight Consistent Trajectory (SCoT) model, a new distillation methodology for generative diffusion models based on a pre-trained teacher model that is purported to enjoy the advantages of rectified flow and consistency model distillation, two leading modes of diffusion model distillation, without their limitations (namely reduced sampling efficiency and error accumulation). SCoT is designed to be fast and to yield straight and consistent trajectories. This is achieved via a carefully designed projection function -- mapping the current value of the trajectory to any future value -- by minimizing a loss function comprised of two terms, one ensuring that the the gradient of the projection mapping remains as constant as possible, the other corresponding to the soft-consistency loss of Kiem at al. (2024), enforcing consistency for the student's trajectories. The good empirical performance of SCoT relative to several other distillation procedures is demonstrated through various numerical experiments on the CIFAR-10 and  ImageNet datasets using various evaluation metrics for efficiency and quality of the sampled images. Though the CMT methodology of Kim et al. (2024) achieves lower FID scores, SCoT is a close second. Further experiments demonstrate the benefits of a two-pronged loss and the use of a loss weighting parameter to achieve an optimal trade-off between the consistency and velocity components of the loss function.

**Questions:**

I have only very minor comments/questions.

- Line 130: there seems to be a typo in the chain of inequalities - shouldn't it be s > t_1 > t_2 (according to eq (3)) or s < t_1 < t_2 (according to the notation introduced on page 2 that t_1 < t_{i+1}?

- The legends of Tables 4 and 5 do not specify that the columns correspond to training iterations. Thts is explained only later, in the legend of table 6.

**Ethical Concerns:**

["NO or VERY MINOR ethics concerns only"]

**Limitations:**

Yes

**Quality:**

3

**Strengths And Weaknesses:**

# Strength
The SCoT distillation procedure for diffusion models is well-motivated and shown empirically to be very competitive. The paper is clearly and carefully written, provides good reference and extensive context for its results. The numerical experiments are carefully designed and compelling.



# Weaknesses
There are no evident weaknesses in the paper or the proposed algorithm. The fact that the CMT methodology of Kim et al. (2024) outperforms SCoT in terms of FID scores should not be considered a weakness.

---

> ### Author Rebuttal · Authors · 2025-07-30
>
> We sincerely thank reviewer **dSXG** for the encouraging feedback of our work and the constructive suggestions. Below we address the two points raised:
>
> >
>
> >**Q1**: Line 130: there seems to be a typo in the chain of inequalities - shouldn't it be s > t_1 > t_2 (according to eq (3)) or s < t_1 < t_2 (according to the notation introduced on page 2 that t_1 < t_{i+1}?
>
>
> >
>
> **[The inequality has been corrected to reflect the correct temporal order as defined in Equation (3).]**
>
> Thank you for pointing this out. The original inequality was inconsistent with the temporal ordering $ s< t_1 < t_2$ described on Page 2. We have revised Line 130 to ensure it aligns with the notation and definitions used throughout the paper.
>
> >
>
> >**Q2**: The legends of Tables 4 and 5 do not specify that the columns correspond to training iterations. Thts is explained only later, in the legend of table 6.
>
> >
>
> **[We have updated the captions of Tables 4 and 5 to clarify that the columns correspond to training iterations.]**
>
> We appreciate the observation. The original legends did not clearly indicate that the columns refer to specific training steps. We have now revised the captions of Tables 4 and 5 to explicitly state that the columns correspond to results at different training iterations consistent with Table 6.

---

### Official Review · Reviewer_nd3F · 2025-07-22

**Clarity:** 3
**Significance:** 3
**Originality:** 2
**Rating:** 4
**Confidence:** 4

**Summary:**

The paper presents SCoT (Straight Consistent Trajectories), a framework for distillation of pre-trained diffusion model for inference with fewer steps.  The approach combines the benefits of Rectified flow, for straightening the trajectory of diffusion models and Consistency models, that ensures that all points along a trajectory map to the same future step. The method is compared to several relevant baselines and state of the art quantitative performance is demonstrated

**Questions:**

1. What is the quantitative performance of generation of higher resolution images from the models that it was currently tested on?
2. If CTM performs about the same, what is the need for using rectified flow for ensuring performance of a few step distilled model?

**Ethical Concerns:**

["NO or VERY MINOR ethics concerns only"]

**Final Justification:**

The rebuttal does a reasonable job of addressing all of the concerns raised. To that end, I am keeping my current score of acceptance.

**Limitations:**

The authors fail to provide an adequate treatment of the limitations of the approach. In particular, the proposed approach has not been validated for high resolution outputs (which has been left for future work). Providing some context about the difficulty of the balancing the losses and the failure modes seen in the outputs of the distilled models would be helpful.

**Paper Formatting Concerns:**

-

**Quality:**

3

**Strengths And Weaknesses:**

## Strengths:

1. **Simple and elegant**: The presented framework is an elegant combination of the best aspects of rectified flow based models and consistency models
2. **Ablation**: The authors provide a number of ablations that justify their design choices.
3. **Supplementary**: The appendix provides a lot of useful information about the the loss design.
4. **Quantitative performance**: The approach shows quantitative performance similar to state of the art CTM model.
5. **Clarity**: The paper is well written with all the necessary details explained adequately.

## Weaknesses:

1. **Novelty**: The novelty of the approach is somewhat limited to the combination of losses from Rectified flow and consistency models. Details regarding the key considerations were needed to make these two losses work together would be insightful.
2. **Qualitative results**: The authors show a number of tables demonstrating quantitative performance, however only include few images for the qualitative results. Including the qualitative performance of CIFAR would be helpful. Furthermore, demonstrating the performance on a more general class of diffusion models would help strengthen the claims of the paper.
3. **Experiments on limited resolution**: The experiments are performed on images upto a resolution of 64x64 where the effect of faster sampling might not be too apparent. In particular, showing performance improvement on larger resolution while still maintaining quality would be beneficial.

---

> ### Author Rebuttal · Authors · 2025-07-30
>
> We thank the reviewer **nd3F** ’s constructive comments and provide our responses below.
>
> >
>
> >**W1**: Novelty: The novelty of the approach is somewhat limited to the combination of losses from Rectified flow and consistency models.
>
> >
>
>
> **[SCoT DOES NOT use the loss from Rectified flow.]** In fact, **our SCoT is the first to discover the connections between CTM and rectified flow, by observing that $\frac{\partial G\_{\phi}(x\_t, t, s)}{\partial s}$ matches to the velocity of $x\_s$.** This discovery is nontrivial. As a result, the velocity loss in SCoT is not the same as the one in rectified flow.
>
>
> >
>
> >**W1.2**: Details regarding the key considerations were needed to make these two losses work together would be insightful.
>
> >
>
> We agree that combining trajectory straightness and consistency requires careful design. These two objectives act on different aspects of the trajectory and are not in conflict: one regularizes the *direction* (velocity), and the other ensures *alignment* across time. To further balance their effects, we introduce a joint objective in Equation (4), with normalization strategies discussed in Section 4.6 to ensure stable and effective optimization. We clarify this rationale further in Appendix A.
>
> The intuition behind this dual-objective design is also visually demonstrated in Figure 1 and theoretically motivated in Appendix A.
>
> >
>
> >**W2**: Qualitative results: The authors show a number of tables demonstrating quantitative performance, however only include few images for the qualitative results. Including the qualitative performance of CIFAR would be helpful. Furthermore, demonstrating the performance on a more general class of diffusion models would help strengthen the claims of the paper.
>
> >
>
> **[Instead, we reported qualitative results on ImageNet in Appendix Figures 3–5 of SCoT, and quantitative results on CIFAR-10 are provided in Table 2.]**
>
>
>
> Due to the **policy prohibiting  updates to the anonymous repository this year** we were unable to include CIFAR-10 visualizations. We chose to show ImageNet-64 (1M samples with $64\times 64$ resolution) samples instead, as the task is more challenging than CIFAR-10 (50K training samples with $32 \times 32$ resolution). Nevertheless, Table 2 shows strong quantitative results on CIFAR-10 across multiple metrics, indicating high-quality generation.
>
>
> >
>
> >**W3**: Experiments on limited resolution: The experiments are performed on images upto a resolution of 64x64 where the effect of faster sampling might not be too apparent. In particular, showing performance improvement on larger resolution while still maintaining quality would be beneficial.
> **Q1**: What is the quantitative performance of generation of higher resolution images from the models that it was currently tested on?
>
> >
>
>
> **[We follow previous protocols to evaluate SCoT, CIFAR-10 and ImageNet-64 provide sufficient validation of model effectiveness, especially given that high resolution data is typically handled by latent diffusion models that compress images to 64×64 in practical applications.]**
>
>
>
> Our experiments are based on widely used benchmarks such as CIFAR-10 and ImageNet-64, following the setup in Consistency Models (ICML23), CTM (ICLR24). These datasets support consistent and reproducible evaluation in low resolution settings, which is the current focus of most trajectory distillation studies.
>
>
>
> Also, for larger datasets such as ImageNet-256, suitable pretrained models with compatible outputs like velocity fields or projection functions are not publicly available. Training such teachers from scratch would require substantial computational resources, which fall outside the current scope of this work.
>
>
>
> Further, many diffusion models can already be observed at $64\times 64$ resolution and adapt to latent diffusion models. In high resolution image synthesis (e.g., $512\times 512$ or $1024\times 1024$), it is common to use VAE encoder in latent diffusion models to compress images into a lower-dimensional latent space, often around $64\times 64$. Therefore, studying models at this resolution is both meaningful and aligned with how large-scale generation is typically implemented.
>
>
>
> >
>
> >**Q2**: If CTM performs about the same, what is the need for using rectified flow for ensuring performance of a few step distilled model?
>
> >
>
> **[The necessity of using velocity loss has been verified in Section 4.4, 4.5, which executes ablation studies for individual components of Equation (2) and Equation (3).]** Specifically, we evaluate the individual impact of the velocity loss and the consistency loss, showing how each contributes to the final performance. The results demonstrates that both velocity loss and consistency loss are essential for achieving fast and high-quality generation.
>
>
> >
>
> >**L1**: The authors fail to provide an adequate treatment of the limitations of the approach. In particular, the proposed approach has not been validated for high resolution outputs (which has been left for future work). Providing some context about the difficulty of the balancing the losses and the failure modes seen in the outputs of the distilled models would be helpful.
>
>
> >
> Please refer to **Reviewer ovkA L1**.

---

### Note · Authors · 2025-08-12

**Dear NeurIPS 2025 Reviewers, AC, SAC, and PC**,

We sincerely thank all the reviewers for their insightful and constructive feedback, which has greatly enhanced our paper on SCoT.

We have addressed key concerns across the board, including novelty and loss integration clarifications (**nd3F**), minor clarifications on notations and tables (**dSXG**), additional baselines, ablations on NFEs beyond 2, individual loss impacts, trajectory straightness measurements, and compatibility with adversarial training (**ovkA**); memory usage, evaluation consistency, and training stability analyses (**Yq6Q**); and comparisons with CTM (without GAN due to stability issues), Diff-Instruct, DMD, sCM, along with flow matching and consistency model connections (**qykR**). These updates, supported by new experiments and discussions on limitations, reinforce SCoT's contributions in unifying consistency models and rectified flows. We commit to integrating these improvements in the final version.

Dear reviewer **ovkA**, can you please check our latest response for the issues of ablations on NFEs beyond 2, individual loss impacts, trajectory straightness. We believe these provided results have fully addressed your concerns. Thanks.

Best,
Authors

---

### Decision · Program_Chairs · 2025-09-17

**Decision:**

Accept (poster)

**Comment:**

Tl;dr: Based on the reviews, rebuttal and ensuing discussion I recommend accept.

### Paper Summary

The paper introduces a new method for distilling pre-trained diffusion models. The primary claim is that SCoT can generate high-quality samples in very few steps by producing trajectories that are both straight and consistent. This is achieved by combining two objectives: a velocity loss that encourages straight trajectories and a consistency loss that ensures that points along the trajectory are mapped to the same final output. Experimental results on CIFAR-10 and ImageNet are presented, with competitive performance compared to SOTA methods, especially in low NFE settings.

### Key strengths and weaknesses

Strengths: 1) Novel and simple method, 2) Good empirical results, particularly on ImageNet, with meaningful ablation studies, 3) Clear presentation, with intuitive figures relating to other approaches.

Weaknesses: 1) Some lingering concerns about novelty among reviewers as a simple combination of existing losses. Though, the authors have clarified that the velocity loss is not identical to the one in rectified flow, 2) Insufficient ablation in the initial draft, those these were addressed during the discussion, 3) Lack of high resolution experiments.

### Decision justification

The proposed method is novel and achieves strong empirical results. The authors have done a good job of addressing the concerns raised by the reviewers during the rebuttal period, providing additional experiments and clarifications. While there are some limitations, e.g. the lack of high resolution experiments, the paper makes a solid contribution to the field and is likely to be of interest to the NeurIPS community.

The discussion period was very active with significant back and forth. The main points raised by the reviewers were: 1) Novelty (R:nd3F), 2) Missing baselines and ablations (R:ovkA, R:Yq6Q, R:qykR), 3) Lack of high resolution experiments (R:nd3F).

The authors provided additional results in the rebuttal, including comparisons with more baselines and ablations on the impact of individual loss components and performance at different NFEs. This has significantly improved the paper and addressed many of the reviewer concerns. Reviewers ovkA and Yq6Q raised their scores after the rebuttal.